# Interfering with Interference: Blind Shuffling and Superposition for Better Multi-Model Compression

## Abstract

We present two complementary random mechanisms to significantly reduce interference when eliminating cross-model redundancy for efficient multi-model serving: *Layer Shuffling* and *Task Vector Superposition*. They work together to increase the orthogonality among interfering task vectors, forcing them into self-destruction without requiring any post-training learning or optimization. *Layer Shuffling* randomly reorders layers of each individual models to reduce the alignment between interfering task vectors. While *Task Vector Superposition* leverages random orthogonal transformations to decorrelate task vectors further. Together, these techniques drastically minimize interference, yielding improved performance across multiple tasks with effectively zero incremental memory cost when incorporating new models. Their data and model-independent nature also allows for seamless on-the-fly addition or removal of models, without requiring any re-computation, making them highly practical for real-world deployment scenarios.

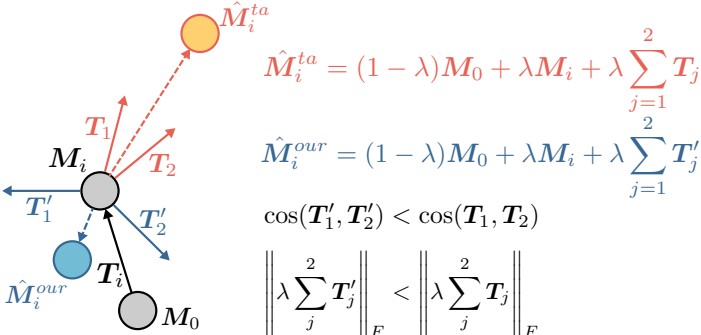

**Figure 1:** Illustration of interference reduction in multi-model compression. $M_0$ is the pre-trained checkpoint, and $M_i$ the $i$-th fine-tuned checkpoint, with task vectors $T_i$, $T_1$, and $T_2$. Standard task arithmetic ($\hat{M}_i^{ta}$, red) sums aligned task vectors, causing interference. With our method ($\hat{M}_i^{our}$, blue), layer shuffling and superposition decorrelate the interfering task vectors into $T_1'$ and $T_2'$, lowering the Frobenius norm of interference. This allows better retrieval of $M_i$ with a higher merging coefficient $\lambda$.

## 1 Introduction

Contemporary advances in machine learning are fueled by ever larger models. Language models and multimodal language models now run into billions of parameters (Cohen & Gokaslan, 2020; Radford et al., 2019; 2021; Workshop et al., 2022). These models are often finetuned into task-specific models to capture the intricacies of individual tasks. As the number of these large models proliferates, serving them becomes a challenge. It is no longer possible to even store multiple models, even in high-end GPUs, significantly impacting downstream applications. Approaches to compress these models without losing accuracy are thus becoming increasingly important. When there are multiple models finetuned from the same pre-trained checkpoint on potentially related tasks, one would expect that the models have a lot of redundant information and can be compressed

together. In fact, a line of work has shown that all these models can be *merged* together into a single model that can tackle all the tasks involved (Ainsworth et al., 2022; Frankle et al., 2020; Wortsman et al., 2020; 2022; Li et al., 2024; Yadav et al., 2024; Tang et al., 2024c; Ilharco et al., 2022; Yang et al., 2023). Of these, a popular framework is task arithmetic (Ilharco et al., 2022), which computes the difference between finetuned model weights and pre-trained model weights to produce task vectors for each task, and add these task vectors together with the pretrained model weights to yield a merged model.

However, the accuracy of these merged models still lags behind the accuracy of the original fine-tuned models. Prior work has identified as a potential reason the interference between the different tasks, which may not be perfectly correlated with each other (Yadav et al., 2024; Wang et al., 2024; Ortiz-Jimenez et al., 2024). While many techniques have been proposed to limit interference, it has generally been difficult to reduce.

In this paper, we take a renewed look at this problem, and find that the cause of this interference is not that the models involved are too different, but *that they are too similar*. Concretely, we find that task arithmetic works best when the task vectors are as orthogonal to each other as possible. Armed with this insight, we propose two new ways of improving upon task arithmetic. Our first approach is to *shuffle* task vectors across the layers of each model before combining them, with an inverse shuffling applied at test time. Our second approach is to apply a random sign flip or reflection to each layer of the task vectors before merging them, again inverting the transformation at test time. Both approaches significantly reduce interference between the task vectors. They also have the advantage of being simple, efficient and requiring no training or optimization.

We test our approach on three different benchmarks involving large models and their finetuned versions: CLIP-ViT-B/32 and CLIP-ViT-L/14 for zero-shot image classification (Radford et al., 2021), Flan-T5-base for text generation (Longpre et al., 2023), and GPT-2 for text classification (Radford et al., 2019). We find that across all of these benchmarks, our approach substantially improves in terms of accuracy over prior model merging-based approaches. When compared to the original fine-tuned models, in two of the three benchmarks our approach yields near-identical accuracy to the individual models while reducing the storage costs by $4\times$. In sum, our contributions are:

1. We provide an analysis of the interference between tasks in task arithmetic, which suggests that similarity between the task vectors may be a problem.

2. We propose two complementary strategies for reducing interference. Our first strategy randomly shuffles parameter matrices across layers. Our second strategy applies a random rotation or a sign flip to the task vectors before merging.

3. We demonstrate through experiments on three benchmarks that our approach compresses multiple models together and achieves much higher accuracy than prior model merging based approaches.

## 2 PROBLEM SETUP

We are given $T$ models $\{\Theta_i\}_{i=1}^T$ fine-tuned from a single pre-trained model $\Theta_0$ on tasks $i = 1, \ldots, T$. Each model $\Theta_i$ is a set of parameter matrices:

$$\Theta_i = \left\{ \mathbb{I}_i, \left( \boldsymbol{M}_i^{k,1}, \boldsymbol{M}_i^{k,2}, \ldots, \boldsymbol{M}_i^{k,m_k} \right)_{k=1}^K, \mathbb{O}_i \right\}.$$

Here $\mathbb{I}_i$ and $\mathbb{O}_i$ are the input and output layers. Each model has $K$ blocks, with the $k$-th block containing $m_k$ matrices $\boldsymbol{M}_i^{k,1}, \boldsymbol{M}_i^{k,2}, \ldots, \boldsymbol{M}_i^{k,m_k}$.

Our goal is to *compress* $\{\Theta_i\}_{i=1}^T$ into a compact representation $\Theta_* = \texttt{compress}(\{\Theta_i\}_{i=1}^T)$ with minimal memory usage, so that at test time, given a task $i$, we can retrieve an *approximate* model $\hat{\Theta}_i = \texttt{retrieve}(\Theta_*, i)$ for the task that achieves high accuracy on this task.

One way to address this problem is obviously to compress each individual model using strategies such as pruning or quantization. However, here we are interested in techniques that can leverage the structure of the problem (namely, $T$ models finetuned from the same source) to yield storage that is *sub-linear* in $T$.

## 3 TASK ARITHMETIC

A promising line of approaches for this problem derives from the observation that models fine-tuned for different tasks can be *merged* into a single model that gives reasonable accuracy for all tasks (Ilharco et al., 2022). Concretely, their framework, called task arithmetic, first computes the difference between the finetuned model weights for each layer $k$, $M_i^k$, and the corresponding pre-trained model weights, $M_0^k$, to produce *task vectors* $T_i^k = M_i^k - M_0^k$. Task arithmetic then computes a weighted average of the pretrained model weights and the task vectors:

$$M_\star^k \leftarrow M_0^k + \lambda \sum_{i=1}^{T} T_i^k = M_0^k + \lambda \sum_{i=1}^{T} (M_i^k - M_0^k), \qquad (1)$$

$$\Theta_\star^{TA} \leftarrow \{M_\star^k\}_{k=1}^{K}, \qquad (2)$$

where $\lambda \in \mathbb{R}^+$ is the merging coefficient. At test time, this compressed model is directly applied no matter what the task:

$$\hat{\Theta}_i \leftarrow \Theta_\star^{TA} = \{M_\star^k\}_{k=1}^{K}. \qquad (3)$$

This approach can be used to reduce model storage by a factor of $T$ since we only need to store one model instead of $T$ different models. However, as we show later, this yields much lower accuracy than the original fine-tuned models.

One reason that has been put forward for the low accuracies offered by task arithmetic is *task interference*: different tasks may want to set particular parameters differently, and merging these parameters naively will cause one task to harm another task's accuracy (Yadav et al., 2024; Wang et al., 2024; Tang et al., 2023). However, a mathematical analysis of this interference is missing. Below, we delve deeper into this interference, and find a counter-intuitive solution.

### 3.1 INTERFERENCE IN TASK ARITHMETIC

To understand the interference term, let us consider what happens when we apply the merged model to task $i$:

$$\hat{M}_i^k = M_\star^k, \qquad \text{(from equation 3)}$$

$$= M_0^k + \lambda \sum_{i=1}^{T} (M_i^k - M_0^k),$$

$$= (1 - \lambda) M_0^k + \lambda M_i^k + \lambda \sum_{j \neq i} T_j^k. \qquad (4)$$

The last line suggests that the model being applied to the $i$-th task is interpolating between the pretrained model $M_0^k$ and the task-specific finetuned model $M_i^k$, but with an additional interference coming from other merged models : $\lambda \sum_{j \neq i} T_j^k$. To retrieve the finetuned model, the first two terms suggest that we should set $\lambda$ to 1. However this will *increase* the interference term, $\lambda \sum_{j \neq i} T_j^k$. Some prior work has tried to achieve a good balance by optimizing $\lambda$ for each model and layer using test-time adaptation(Yang et al., 2023), but attaining this balance has often been challenging.

Instead of focusing on $\lambda$, let us look at this interference term in greater detail by analyzing its Frobenius norm:

$$\left\| \lambda \sum_{j \neq i} T_j^k \right\|_F^2 = \lambda^2 \left( \sum_{j \neq i} \|T_i^k\|_F^2 + 2 \sum_{\substack{1 \leq l < j \leq n \\ l, j \neq i}} \|T_l^k\|_F \|T_j^k\|_F \cos(T_l^k, T_j^k) \right). \qquad (5)$$

We observe that the interference term is directly correlated with two quantities: the magnitude of the task vectors $T_i^k$ (which is out of our control since it depends on the task-specific finetuning), and the cosine products between them. Interestingly, the interference term is maximum when the task

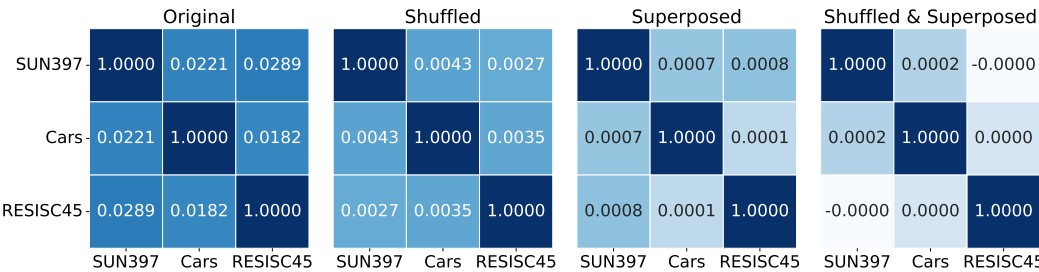

**Figure 2:** Average pairwise cosine similarity of three out of eight CLIP-ViT-B/32 task vectors during model retrieval for SUN397 across three repetitions. Both random layer shuffling and superposition increase mutual orthogonality, with an additive effect when combined.

vectors are very closely aligned with each other. Thus, the problem with task arithmetic is not that the individual task vectors are very different from each other, but that they are *too similar*.

Our goal, therefore, should be to make the task vectors as different from each other as possible. Below, we propose two strategies for doing this.

## 4 METHODOLOGY

As described above, to minimize interference, we want the task vectors to be as orthogonal to each other as possible. We propose two complementary random algorithms to achieve this: *random layer shuffling* and *task vector superposition*.

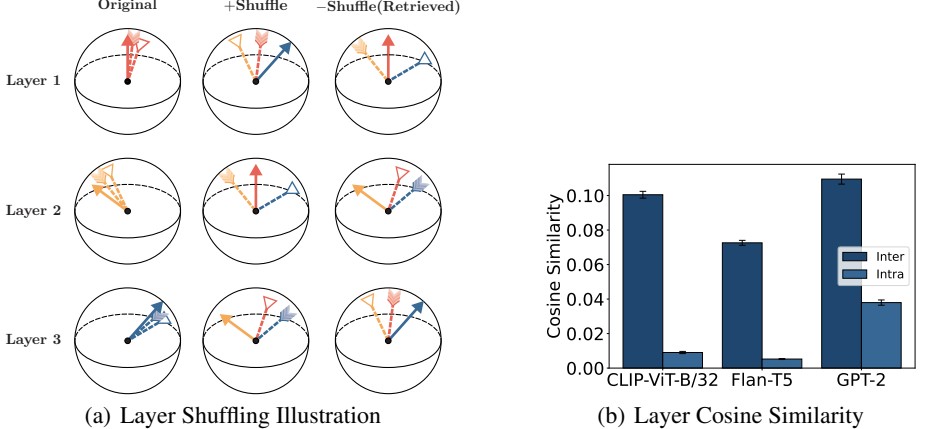

(a) Layer Shuffling Illustration            (b) Layer Cosine Similarity

**Figure 3:** (a) Layer shuffling illustration in a three-layer model with three checkpoints. Different task vectors (distinct arrowheads) initially align across layers, causing interference when directly merged. Shuffling layers followed by inverse transformation retain the target task's orientation (standard arrowhead) while reducing interference through increased orthogonality among other vectors. (b) shows cosine similarity distributions between task vector layers within and across models for CLIP-ViT-B/32, Flan-T5, and GPT-2, with standard error of mean (SEM) as error bars.

### 4.1 RANDOM LAYER SHUFFLING

Across several model architectures (CLIP-ViT-B/32, CLIP-ViT-L/14 (Radford et al., 2021), Flan-T5 (Longpre et al., 2023), and GPT-2 (Radford et al., 2019)), we observed that task vector layers within the same model exhibit greater variability compared to corresponding layers across fine-tuned models (see Figure 3 (b)). This insight suggests that by randomly shuffling layers across different task vectors, we can reduce the pairwise cosine similarity of interfering task vectors and thus minimize their contribution to the interference. To that end, we propose *random layer shuffling* as a simple fix to the problem.

**Method Description.** The models we consider are made of multiple parameter blocks of similar structure. For example, transformers have multiple MLP layers and multiple attention layers. Many of the MLP and Attention layers have weight matrices of the same size.

Our proposal is that when merging the task vectors, for each model, we first *randomly permute* the task vectors across layers of the same type and with the same dimensions of parameter matrices (as illustrated in Figure 3 (a)). Then, when we want to perform inference on a particular task, we perform the inverse of the corresponding permutation to obtain the model for the task.

Concretely, for each task $i$, we produce a random permutation of the layers $\sigma_i$, taking care to only permute across layers of the same type and same dimensionality. We then produce merged task vectors by adding up these shuffled task vectors across models. The merged task vector for the $k$-th layer is:

$$\boldsymbol{T}_\star^k \leftarrow \sum_{i=1}^T \boldsymbol{T}_i^{\sigma_i(k)}. \tag{6}$$

The number of such merged task vectors is equal to the total number of layers $K$. We then store both the pretrained model $\boldsymbol{\Theta}_0$ and the merged task vectors $\{\boldsymbol{T}_\star^k\}_{k=1}^K$:

$$\boldsymbol{\Theta}_*^{Shuffle} \leftarrow \left(\boldsymbol{\Theta}_0, \{\boldsymbol{T}_\star^k\}_{k=1}^K\right). \tag{7}$$

**Reduction in Interference:** Because parameter vectors from different layers are less likely to align, we effectively reduce the cosine product between the task vectors being merged: instead of the term $\cos(\boldsymbol{T}_i^k, \boldsymbol{T}_j^k)$ in the interference term (equation 5), we now have the product $\cos(\boldsymbol{T}_i^{\sigma_i(k)}, \boldsymbol{T}_j^{\sigma_j(k)})$ , which is expected to be significantly lower due to the reduced alignment of parameter vectors from different layers. We thus expect smaller interference and thus more faithful retrieval of model weights for each task. The first two parts of Figure 2 shows this effect in action, with layer shuffling reducing the pairwise cosine similarity among interfering vectors unanimously.

## 4.2 TASK VECTOR SUPERPOSITION

We can also leverage the *blessing of dimensionality* (Gorban & Tyukin, 2018) to promote orthogonality among high dimensional vectors. We take inspiration from Cheung et al. (2019) on continual learning and introduce *superposition* as a complementary approach to increase the mutual orthogonality among interfering task vectors.

**Method Description.** Considering merging the parameters of layer $k$, we sample random binary diagonal matrices whose diagonal entries have equal probability to be $+1$ or $-1$ to each of the $T$ task vectors and apply them to the vectors before summation:

$$\boldsymbol{T}_\star^k \leftarrow \sum_{i=1}^T \boldsymbol{T}_i^k \boldsymbol{C}_i^k. \tag{8}$$

We call them context matrices and $\forall i \in [1, \ldots, T]$, $\boldsymbol{C}_i^k \boldsymbol{C}_i^{k(T)} = \boldsymbol{C}_i^k \boldsymbol{C}_i^{k(-1)} = \boldsymbol{I}$.

When performing task $i$, we apply the inverse transformation $\boldsymbol{C}_i^{k(-1)}$ to retrieve task vector $\boldsymbol{T}_i^k$ from the superposition:

$$\hat{\boldsymbol{T}}_i^k = \boldsymbol{T}_\star^k \boldsymbol{C}_i^{k(-1)}, \tag{9}$$

$$= \sum_{i=1}^T [\boldsymbol{T}_i^k \boldsymbol{C}_i^k] \boldsymbol{C}_i^{k(-1)}, \tag{10}$$

$$= \boldsymbol{T}_i^k + \sum_{j \neq i} [\boldsymbol{T}_j^k \boldsymbol{C}_j^k \boldsymbol{C}_i^{k(-1)}]. \tag{11}$$

We store both the pretrained model $\boldsymbol{\Theta}_0$, the merged task vectors $\{\boldsymbol{T}_\star^k\}_{k=1}^K$, as well as the context matrices $\{\boldsymbol{C}_\star^k\}_{k=1}^K$:

$$\boldsymbol{\Theta}_*^{Superpose} \leftarrow \left(\boldsymbol{\Theta}_0, \{\boldsymbol{T}_\star^k\}_{k=1}^K, \{\boldsymbol{C}_\star^k\}_{k=1}^K\right). \tag{12}$$

**Reduction in Interference.** Two random vectors in high dimensional space is very likely to be nearly orthogonal with each other. Now the cosine similarity in equation 5 changes from $\cos(\boldsymbol{T}_i^k, \boldsymbol{T}_j^k)$ to $\cos(\boldsymbol{T}_i^k \boldsymbol{C}_i \boldsymbol{C}_j^{l(-1)}, \boldsymbol{T}_j^k \boldsymbol{C}_i \boldsymbol{C}_j^{l(-1)})$ when performing task $l$. The randomly sampled diagonal binary matrices $\{\boldsymbol{C}_\star^k\}_{k=1}^K$ will randomize the task vectors, leading to more orthogonal task vectors, smaller interference, and thus better retrieval of model parameters for the task at hand. Again, Figure 2 confirmed the cosine similarity reduction when task vectors are superposed together.

## 5 EXPERIMENTS

We evaluate our methods on FusionBench (Tang et al., 2024a) across vision and language tasks, showing comparable performance for both discriminative and generative models. Through ablation studies, we analyze component importance, merging coefficient effects, and context matrix designs. Finally, we demonstrate broader applications including PEFT model compression and large-scale merging of twenty CLIP-ViT-L/14 models.

### 5.1 EXPERIMENT SETUP

**Datasets and Models.** We follow Tang et al. (2024a) and select three representative scenarios to evaluate our methods. This includes i) CLIP-ViT-B/32 fine-tuned on eight image classification datasets (adopted from (Ilharco et al., 2022)); ii) Flan-T5-base fine-tuned on eight text generation datasets; and iii) GPT-2 fine-tuned on seven text classification datasets. Detailed information on the datasets and models is in Appendix B.

**Baselines and Metrics.** We evaluate baselines from model merging/compression literature, grouped by memory requirements: methods using the original footprint (pre-trained model, standard merging techniques) and those requiring additional memory (fine-tuned models, newer merging baselines). The pre-trained and fine-tuned models provide lower and upper performance bounds respectively. Following Ilharco et al. (2022), we optimize merging coefficient $\lambda$ via validation set grid search. We report accuracy and memory usage across three runs per experiment with random operations. See Appendix B for details.

### 5.2 PERFORMANCE ANALYSIS

**Table 1:** Performance and memory comparison of CLIP-ViT-B/32 models across eight image classification tasks, showing absolute and normalized accuracy (%), as well as memory footprint (Gb). Results averaged over three runs where applicable. Variances smaller than 0.1% are omitted.

| Method | Avg.(%) ↑ | Bits(Gb) ↓ | SUN397 | Cars | RESISC45 | EuroSAT | SVHN | GTSRB | MNIST | DTD |
|---|---|---|---|---|---|---|---|---|---|---|
| Pre-trained | 48.2 (53.4) | 0.564 (1.00) | 63.2 | 59.8 | 60.7 | 46.0 | 31.6 | 32.5 | 48.3 | 43.9 |
| Weight Averaging | 66.5 (73.6) | 0.564 (1.00) | 65.4 | 62.6 | 70.8 | 76.9 | 64.5 | 54.9 | 86.3 | 50.9 |
| Fisher Merging | 70.6 (78.2) | 0.564 (1.00) | 66.7 | 64.0 | 72.2 | 91.6 | 69.0 | 64.3 | 83.5 | 53.7 |
| RegMean | 80.5 (89.1) | 0.564 (1.00) | 67.8 | 68.9 | 82.5 | 94.4 | 90.6 | 79.2 | 94.7 | 63.2 |
| Task Arithmetic | 69.8 (77.2) | 0.564 (1.00) | 64.4 | 64.1 | 70.5 | 80.4 | 73.9 | 62.8 | 93.0 | 51.6 |
| Ties-Merging | 72.2 (80.0) | 0.564 (1.00) | 67.1 | 64.2 | 74.1 | 91.6 | 77.7 | 69.4 | 94.1 | 54.0 |
| Layerwise AdaMerging | 82.6 (91.5) | 0.564 (1.00) | 67.9 | 71.3 | 83.5 | 92.7 | 87.4 | 92.9 | 98.2 | 67.0 |
| PSP | 4.5 (5.0) | 0.564 (1.00) | 0.3 | 0.5 | 1.9 | 10.4 | 8.8 | 2.3 | 9.8 | 1.9 |
| Fine-tuned | 90.3 (100) | 2.84 (5.03)[1] | 75.0 | 78.3 | 95.2 | 99.0 | 97.3 | 98.9 | 99.6 | 79.7 |
| WEMoE | 89.2 (98.8) | 2.27 (4.03) | 73.7 | 76.8 | 93.4 | 98.2 | 96.8 | 98.2 | **99.6** | 76.6 |
| SMILE | 89.3 (98.9) | 1.23 (2.20) | 73.6 | **77.8** | 92.0 | 98.3 | 96.9 | 98.1 | **99.6** | 78.1 |
| **TA+Shuffle (Ours)** | 81.3 (90.0) | 0.89 (1.58) | 65.6 | 58.5 | 86.8 | 94.5 | 93.2 | 91.4 | 98.5 | 62.2 |
| **STA (Ours)** | 89.6 (99.2) | 0.89 (1.58) | 74.4 | 75.6 | 94.6 | **99.0** | 97.1 | 98.5 | 99.5 | 77.8 |
| **STA+Shuffle (Ours)** | **89.9** (99.6) | 0.89 (1.58) | **74.8** | 76.7 | **94.8** | **99.0** | **97.2** | **98.6** | 99.5 | **78.7** |

**Superior MTL Performance.** Our approach achieves significant accuracy gains across benchmarks (Tables 1, 2 and 6), with *STA+Shuffle* nearly matching individual fine-tuned models. We outperform WEMoE (Tang et al., 2024c) and SMILE (Tang et al., 2024b) on image classification while using only 40% and 72% of their respective memory footprints, and surpass SMILE's text generation performance at benchmark saturation. Though Task Arithmetic (Ilharco et al., 2022) uses 55% of our storage, its performance is substantially lower. Parameter Superposition (Cheung et al.,

**Table 2:** Performance and memory comparison of Flan-T5-base models across eight GLUE text generation tasks, showing absolute and normalized accuracy (%), as well as memory footprint (Gb). Results averaged over three runs where applicable. Variances smaller than 0.1% are omitted.

| Method | Avg.(%) ↑ | Bits(Gb) ↓ | CoLA | MNLI | MRPC | QNLI | QQP | RTE | SST2 | STSB |
|---|---|---|---|---|---|---|---|---|---|---|
| Pre-trained | 75.7 (87.6) | 1.19 (1.00) | 69.1 | 56.5 | 76.2 | 88.4 | 82.1 | 80.1 | 91.2 | 62.2 |
| Weight Averaging | 78.9 (91.3) | 1.19 (1.00) | 69.1 | 62.6 | 79.4 | 89.8 | 83.9 | 81.2 | 91.7 | 73.2 |
| Task Arithmetic | 79.6 (92.1) | 1.19 (1.00) | 69.7 | 64.1 | 79.2 | 90.2 | 83.9 | 81.6 | 92.1 | 76.4 |
| Ties-Merging | 79.9 (92.5) | 1.19 (1.00) | 70.3 | 65.0 | 78.9 | 90.2 | 83.5 | 81.6 | 91.7 | 78.3 |
| PSP | 0.0 (0.0) | 1.19 (1.00) | 0.0 | 0.0 | 0.0 | 0.0 | 0.0 | 0.0 | 0.0 | N/A |
| Fine-tuned | 86.4 (100) | 9.52 (8.00) | 75.0 | 83.4 | 87.5 | 91.5 | 85.4 | 85.9 | 93.6 | 88.7 |
| SMILE | 85.5 (99.0) | 1.81 (1.52) | 73.2 | **84.2** | 85.0 | 91.3 | 84.9 | 84.8 | 93.5 | 87.3 |
| **TA+Shuffle (Ours)** | 85.7 (99.0) | 2.38 (2.00) | 75.5 | 82.0 | 87.5 | 91.1 | 83.9 | 83.8 | **93.6** | 88.4 |
| **STA (Ours)** | **86.5** (100) | 2.38 (2.00) | **77.2** | 82.1 | 87.6 | 91.6 | **85.3** | **85.7** | 93.2 | **89.0** |
| **STA+Shuffle (Ours)** | 86.4 (100) | 2.38 (2.00) | 75.6 | 82.8 | **88.2** | **91.7** | **85.3** | **85.7** | 93.5 | 88.9 |

2019), while effective for continual learning, underperforms here, demonstrating the importance of task vector superposition for offline compression.

**Amortizable Memory Overhead.** Our method requires only 2x memory, mainly from storing merged task vectors $\{T_\star^k\}_{k=1}^K$ and binary context matrices $\{C_\star^k\}_{k=1}^K$ (equations 7, 12). By storing random seeds and regenerating context matrices on-the-fly with minimal overhead (292.70 ms for CLIP-ViT-B/32, 658.19 ms for CLIP-ViT-L/14 on Intel Xeon Gold 6448Y CPU), we achieve effectively zero additional memory per model. This enables efficient scaling, demonstrated by merging 20 CLIP-ViT-L/14 models with state-of-the-art performance and 9x memory reduction (Sec. 5.9).

## 5.3 KEY COMPONENTS ABLATION

In this section, we ablate both the *random layer shuffling* and *superposition* to show their individual contribution. Specifically, we derive two variants:

- **TA+Shuffle**: we randomly shuffle the layers before task arithmetic without performing superposition.

- **STA**: we superpose task vectors without layer shuffling.

As shown in Tables 1, 2, 4, 5 and 6, both methods significantly outperform task arithmetic consistently. In some benchmarks, shuffling works better (Flan-T5-base and GPT-2) while superposition works better in others (CLIP). This difference may be because of the nature of task vectors themselves: how they vary across the model layers and across different models. We find that the combined approach is able to combine gains from both components, yielding consistently the best result across all the benchmarks. This complementary effect is further manifested in Figure 2, where introducing both mechanism gets the smallest pairwise cosine similarity among interfering task vectors. We provide a more detailed analysis of the interplay between shuffling and superposition in Section D.

## 5.4 IMPACT OF MERGING COEFFICIENT $\lambda$

Here we examine the interplay between the merging coefficient $\lambda$ and the average performance across different setup. For each variant derived in section 4, we perform a grid search on $\lambda = \{0.1, 0.2, \cdots, 1.0\}$ when compressing eight CLIP-ViT-B/32 models for image classification. Figure 4 shows the change of optimal model performance and the coefficient $\lambda$ when *layer shuffling* and *superposition* are introduced to task arithmetic.

We observe that when shuffling and superposition are introduced, the best performance increases along with the value of $\lambda$. This shows the effectiveness of our method in reducing interference, allowing larger $\lambda$ to be selected for more authentic model retrieval as according to equation 4.

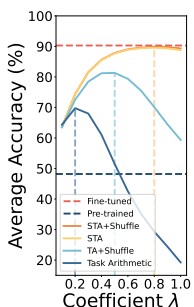

**Figure 4:** The impact of $\lambda$ on average accuracy over eight image classification tasks.

## 5.5 IMPACT OF CONTEXT MATRIX DESIGN

To further examine how *task vector superposition* works and shad light on better context matrix design, we make a comparison between three types of context matrices: random binary diagonal matrix with $\{-1, +1\}$ entries (RBD), identity matrix (Identity), and random diagonal matrix with entries draw from Normal distribution (RD). We use random layer shuffling when compressing the 8 CLIP-ViT-B/32 models on the image classification tasks.

The average accuracy and its variance with the optimal merging coefficient is shown in Figure 5. RBD receives higher accuracy than Identity due to the randomness it introduces, which reduces interference as discussed in section 4.2. Despite being random, RD's accuracy is much lower than RBD. We think this happens because RD is not an orthogonal matrix. It fails to preserve the Frobenius norm of $\boldsymbol{T}_j^k \boldsymbol{C}_j^k \boldsymbol{C}_i^{k(-1)}$ and thus disturb this self-cancellation process.

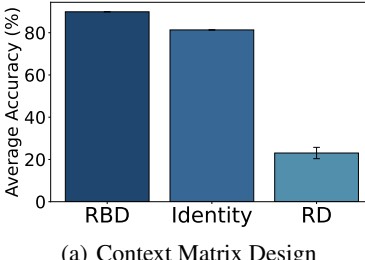 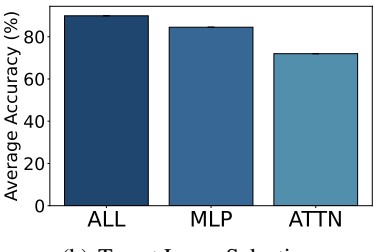

| (a) Context Matrix Design | (b) Target Layer Selection |
|---|---|

**Figure 5:** (a) Impact of context matrix design to the average accuracy. RBD stands for random binary diagonal; RD stands for random diagonal. (b) Impact of target layer selection to the average accuracy. ALL stands for choosing all layers; MLP means only MLP layers are selected; and ATTN stands for attention layers.

## 5.6 TARGET LAYER SELECTION

By default we apply the random operations on all layers within the models. In this section, we evaluate the benefits of targeting specific types of layers. To do this, we create two variants: MLP (which selects only the MLP layers) and ATTN (which selects only the attention layers), in addition to the default setup (ALL). Figure 5 shows the average accuracy for each setup across eight image classification tasks using CLIP-ViT-B/32. The ALL configuration achieves the highest accuracy, followed by MLP and ATTN. Note that the total number of parameters in MLP is twice that of ATTN, explaining the gradual decline in performance as fewer parameters are selected.

## 5.7 MODEL HOT SWAPPING

The ability to hot-swap models in real-world applications is crucial, especially in dynamic environments like model serving, where new models need to be integrated into the system regularly, and deprecated ones need to be removed in a timely fashion. As mentioned, the *STA+Shuffle* method allows for this by shuffling layers and sampling diagonal binary matrices *independently* of data or model parameters, thus enabling the on-the-fly addition of new models without the need for recomputation. This provides a big advantage over methods like WEMoE which require recomputation of the router when new models are added (Tang et al., 2024c), or TALL-masks, which needs to recompute all task masks when a new model is added (Wang et al., 2024).

**Table 3:** Comparison of selected methods with hot adding and recomputation requirements when new models are added to the pool.

| Method | Hot Swap | Recomputation |
|---|---|---|
| Task Arithmetic | ✓ | ✗ |
| WEMoE | ✗ | ✓ |
| TALL-masks | ✗ | ✓ |
| STA+Shuffle | ✓ | ✗ |

## 5.8 PARAMETER EFFICIENT FINETUNING (PEFT) MODEL COMPRESSION

We can also apply our method on PEFT adapter weights. Consider LoRA (Hu et al., 2021), where we have a fixed pre-trained model $\Theta_0$, along with LoRA weights $L_i$. We merge the LoRA weights to get the fine-tuned model: $\Theta_i = \Theta_0 + \lambda L_i$. Similar to section 4.1 and 4.2, we apply *random layer shuffling* and *superposition* on these LoRA weight vectors before retrieval.

Experiments on Flan-T5-base LoRA fine-tunes (Longpre et al., 2023; Tang et al., 2024a;b) demonstrate that our method is performative in PEFT compression settings as well (Table 4). With 99.8% normalized average accuracy compare to the fine-tuned baseline, and 1.20 Gb memory usage, our method presents a better trade-off point between performance and storage usage than the state-of-the-art model SMILE (Tang et al., 2024b).

**Table 4:** Performance and memory comparison of Flan-T5-base LoRA models across eight GLUE text generation tasks, showing absolute and normalized accuracy (%), as well as memory footprint (Gb). Results averaged over three runs where applicable. Variances smaller than 0.1% are omitted.

| Method | Avg.(%) ↑ | Bits(Gb) ↓ | CoLA | MNLI | MRPC | QNLI | QQP | RTE | SST2 | STSB |
|---|---|---|---|---|---|---|---|---|---|---|
| Pre-trained | 75.7 (87.6) | 1.19 (1.00) | 69.1 | 56.5 | 76.2 | 88.4 | 82.1 | 80.1 | 91.2 | 62.2 |
| Weight Averaging | 78.2 (92.4) | 1.19 (1.00) | **69.7** | 59.7 | 78.9 | 90.1 | 83.8 | 80.5 | 91.2 | 72.0 |
| Task Arithmetic | 77.4 (91.5) | 1.19 (1.00) | 68.8 | 55.2 | 78.7 | 89.8 | 83.7 | 79.1 | 91.5 | 72.4 |
| Ties-Merging | 77.5 (91.6) | 1.19 (1.00) | 68.3 | 56.3 | 79.4 | 89.8 | 83.7 | 79.4 | 91.6 | 71.2 |
| Fine-tuned | 84.6 (100) | 1.25 (1.05) | 69.1 | 82.7 | 85.5 | 90.9 | 84.0 | 84.4 | 92.9 | 87.4 |
| SMILE | 84.0 (99.3) | 1.21 (1.02) | 69.3 | **82.9** | 83.8 | 90.6 | 83.9 | 83.4 | **93.1** | 85.1 |
| **TA+Shuffle (Ours)** | 83.9 (99.2) | 1.20 (1.01) | 69.2 | 79.0 ±0.3 | 84.2 | 90.4 | **84.1** | **85.0** | 92.9 | 86.5 |
| **STA (Ours)** | 83.0 (98.1) | 1.20 (1.01) | 69.1 | 81.3 | 82.2 | 90.5 | 83.2 | 79.1 | 92.7 | 85.6 |
| **STA+Shuffle (Ours)** | 84.4 (99.8) | 1.20 (1.01) | 69.1 | 82.7 | **85.0** | 90.9 | 83.8 | 84.2 | 92.7 | **86.9** |

## 5.9 SCALABILITY ANALYSIS

Our method scales effectively to merging larger models and more tasks, as demonstrated on CLIP ViT-L/14 with 8, 14, and 20 image classification tasks (Table 5). STA+Shuffle achieves near fine-tuned performance (93.5% vs 94.2% for 20 tasks) while maintaining constant 2.87GB storage regardless of task count. In contrast, the state-of-the-art model TALL-masks+TA (Wang et al., 2024) requires progressively more storage (5.42GB to 9.25GB) as tasks increase. Though Task Arithmetic uses only 1.59GB, its performance drops significantly with more tasks. Model retrieval remains efficient, requiring just 658.19ms per CLIP ViT-L/14 model on an Intel Xeon Gold 6448Y CPU.

**Table 5:** Performance and memory comparison of CLIP ViT-L/14 models across three test scenarios with 8, 14, and 20 image classification tasks, showing absolute and normalized accuracy (%), as well as memory footprint (Gb). Results averaged over three runs where applicable. Variances smaller than 0.1% are omitted.

| Method | 8 tasks | | 14 tasks | | 20 tasks | |
|---|---|---|---|---|---|---|
| | Acc.(%) ↑ | Bits(Gb) ↓ | Acc.(%) ↑ | Bits(Gb) ↓ | Acc.(%) ↑ | Bits(Gb) ↓ |
| Pre-trained | 64.5 (68.3) | 1.59 (1.00) | 68.1 (72.8) | 1.59 (1.00) | 65.2 (69.2) | 1.59 (1.00) |
| Task Arithmetic | 84.0 (88.7) | **1.59 (1.00)** | 79.1 (84.2) | **1.59 (1.00)** | 73.8 (78.3) | **1.59 (1.00)** |
| Fine-tuned | 94.4 (100) | 10.53 (6.62) | 93.5 (100) | 18.18 (11.43) | 94.2 (100) | 25.84 (16.25) |
| Magnitude Masking | 92.8 (98.2) | 5.42 (3.41) | 90.6 (96.7) | 7.34 (4.62) | 90.9 (96.4) | 9.25 (5.82) |
| TALL Mask+TA | 94.2 (99.7) | 5.42 (3.41) | 92.4 (98.8) | 7.34 (4.62) | 93.2 (98.9) | 9.25 (5.82) |
| **TA+Shuffle (Ours)** | 93.0 (98.4) | 2.87 (1.81) | 88.8 (94.6) | 2.87 (1.81) | 87.1 ±0.1(92.2) | 2.87 (1.81) |
| **STA (Ours)** | 94.2 (99.8) | 2.87 (1.81) | 92.8 (99.2) | 2.87 (1.81) | 93.4 (99.1) | 2.87 (1.81) |
| **STA+Shuffle (Ours)** | **94.3 (99.9)** | 2.87 (1.81) | **93.0 (99.5)** | 2.87 (1.81) | **93.5 (99.3)** | 2.87 (1.81) |

---

[1]CLIP models' text encoder is frozen and shared by all fine-tuned models.

## 6 RELATED WORK

**Model Merging.** Recent research on model merging is largely founded on *linear mode connectivity (LMC)* (Frankle et al., 2020; Neyshabur et al., 2020), which posits that models fine-tuned from the same pre-trained model are connected by a linear path along which performance remains constant. Building upon this concept, Wortsman et al. (2022) and Li et al. (2024) demonstrated that a set of specialist models can be directly interpolated to obtain a multi-task model. Ilharco et al. (2022) proposed interpolating the parameter deltas (referred to as "task vectors") instead. However, these methods suffer from *task interference*: when different models adjust the same parameters in conflicting ways, summing these adjustments leads to interference and degraded performance on individual tasks (Yadav et al., 2024; Tang et al., 2024b; Wang et al., 2024). To mitigate this interference, various strategies have been proposed. Yang et al. (2023) optimized the merging coefficients for different tasks and layers to reduce interference. Yadav et al. (2024) addressed the conflict by removing redundant parameters and resolving sign disagreements. Tang et al. (2024c) reduced interference by upscaling the multilayer perceptron (MLP) layers. Tang et al. (2024b) compressed task vectors using singular value decomposition (SVD) and performed routing between them to further diminish interference. Both Wang et al. (2024) and Yu et al. (2024) sparsified the task vectors to prevent task conflicts. Additionally, Ortiz-Jimenez et al. (2024) proposed fine-tuning the linearized model along the tangent space of the pre-trained model to promote weight disentanglement and avoid interference. In contrast to the above mentioned methods that aim to avoid conflicts, we intentionally accumulate interference among conflicting task vectors to facilitate their mutual cancellation.

**Model Compression.** Model compression techniques aim to reduce the memory footprint of models while maintaining their performance. Model pruning compresses neural networks by removing inessential parameters in either a structured (Anwar et al., 2017; Fang et al., 2023; He & Xiao, 2023; Wang et al., 2019) or unstructured (Liao et al., 2023; Kwon et al., 2020) manner. Parameter quantization saves memory and speeds up inference by converting the weights and activation values of a neural network from high precision to low precision (Gholami et al., 2022; Liu et al., 2021; Yuan et al., 2022). Knowledge distillation reduces the memory footprint by training a smaller network to mimic a larger network's behavior (Gou et al., 2021; Cho & Hariharan, 2019; Park et al., 2019; Zhao et al., 2022). Leveraging the low-rank nature of model parameters, many works decompose weight matrices into low-rank matrices for memory reduction (Yu et al., 2017; Li et al., 2023; Guo et al., 2024). Ryu et al. (2023) observed the low-rank nature of weight residuals in overparameterized models and proposed reducing storage demands for fine-tuned models through low-rank approximation of these residuals. Similarly, Tang et al. (2024b) compresses individual task vectors using SVD and routes through a set of them conditioned on input. Our work differs from these works in that we try to reduce redundancy across a set of aligned models rather than within them.

## 7 DISCUSSION AND FUTURE WORK

In this work, we introduce *random layer shuffling* and *task vector superposition* to enhance orthogonality between task vectors, thereby significantly reduce task interference during multi-model merging and compression. These data- and model-agnostic random operations enable users to i) efficiently modify the model merging combinations without the need for additional training or optimization; ii) merge additional models without increasing memory usage by saving random seeds. Evaluation on diverse model and task sets demonstrates that our method maintains high performance while keeping a constant memory footprint as more and larger models are merged. These attributes make our approach highly practical for real-world multi-model serving environments.

An interesting future direction is to further improve performance by increasing orthogonality, potentially through alternative random operations or more systematic approaches. Our method relies on specific properties of model parameters that emerge from fine-tuning. Identifying these properties and enhancing fine-tuning strategies could lead to better merging and compression performance. Since we reduce cross-model redundancy, applying model compression algorithms could potentially further decrease memory footprint.

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

## A OVERVIEW

In this appendix we present more information about the experiment settings and analysis that could not fit in the main paper. In Sec. B we present more details about the datasets, models, and baseline methods used in evaluation. In Sec. C we derive the squared Frobenius norm of the interference term used in equation 5. In Sec. D we include additional analysis and results.

## B EXPERIMENT SETUP

This section provides detailed descriptions for the datasets, baselines, and model fine-tuning settings.

**Datasets Details.** Evaluation are performed on two sets of datasets with different type of tasks.

1. **Image Classification Datasets**: For image classification, following Ilharco et al. (2022); Tang et al. (2024a); Wang et al. (2024), we use twenty tasks from CLIP's (Radford et al., 2021) test set: SUN397 (Xiao et al., 2010), Cars (Krause et al., 2013), RESISC45 (Cheng et al., 2017), EuroSAT (Helber et al., 2019), SVHN (Netzer et al., 2011), GTSRB (Stallkamp et al., 2012), MNIST (Deng, 2012), DTD (Cimpoi et al., 2014), CIFAR100 (Krizhevsky, 2009), STL10 (Coates et al., 2011), Flowers102 (Nilsback & Zisserman, 2008), OxfordIIITPet (Parkhi et al., 2012), PCAM (Veeling et al., 2018), FER2013 (Goodfellow et al., 2013), EMNIST (Cohen et al., 2017), CIFAR10 (Krizhevsky, 2009), Food101 (Bossard et al., 2014), FashionMNIST (Xiao et al., 2017), RenderedSST2 (Socher et al., 2013; Radford et al., 2019), and KMNIST (Clanuwat et al., 2018). For experiments on K tasks, the first K datasets from this list are select.

2. **Text Classification and Generation Datasets**: For text classification and generation, following Tang et al. (2024a), we have in total eight tasks from the GLUE benchmark (Wang, 2018): CoLA, MNLI, MRPC, QNLI, QQP, RTE, SST2, and STSB.

**Baseline Details.** Our experiments compare the following baselines and our methods:

- **Pre-trained**: Pre-trained model used across all tasks (performance lower bound).
- **Fine-tuned**: Individual fine-tuned models (performance upper bound).
- **Weight Averaging** (Wortsman et al., 2022): Merge models by directly averaging their parameters.
- **Fisher Merging** (Matena & Raffel, 2022): Fisher Merging uses the Fisher information as a weight for each parameter during weight averaging.
- **RegMean** (Jin et al., 2022): RegMean introduces a constraint in model merging by minimizing the L2 distance between the merged model and each individual model.
- **Task Arithmetic** (Ilharco et al., 2022): Task Arithmetic computes the delta parameters between fine-tuned models and the base model (known as "task vectors") and aggregates them before adding into a pre-trained model.
- **Ties-Merging** (Yadav et al., 2024): Ties-Merging addresses task conflict issues found in Task Arithmetic by eliminating redundant parameters and resolving symbol conflicts.
- **Layer-wise AdaMerging** (Yang et al., 2023): Layer-wise AdaMerging finds optimal merging coefficients for each layer of each task vector in Task Arithmetic using test-time adaptation.
- **Parameter Superposition (PSP)** (Cheung et al., 2019): PSP applies random orthogonal matrices periodically during training to store many models into one model. We adopted it in our offline setting by treating fine-tuned models as different model instances during training to provide some contexts to our task vector superposition approach.
- **WEMoE** (Tang et al., 2024c): WEMoE only merges the layer norm and attention layers while keeping the multi-layer perceptron layers unmerged, with a router to dynamically allocate weights to each MLP conditioned on the input.

- **SMILE** (Tang et al., 2024b): SMILE compresses task vectors with singular value decomposition (SVD). It then determines the routing weights based on the alignment between input and each low-rank matrix.

- **TALL-masks+TA** (Wang et al., 2024): TALL-masks+TA finds a binary parameter mask for each task vector by finding task-specific parameters with values deviate a lot from the aggregated multi-task vector. The corresponding mask for each task is applied to the multi-task vector before adding to a pre-trained model.

- **Magnitude Masking** (Wang et al., 2024): Magnitude Masking differ from TALL-masks in that it determines per-task masks by keeping the top k% of each task vector's parameters.

- **TA+Shuffle (Ours)**: TA+Shuffle performs random layer shuffling among the repetitive layers in each task vector before merging them with Task Arithmetic.

- **STA (Ours)**: STA applies random orthogonal transformations to each layer in each task vector in Task Arithmetic.

- **STA+Shuffle (Ours)**: STA+Shuffle combines layer shuffling and superposition.

**Model Details.** We utilize fine-tuned models from Tang et al. (2024a) and Wang et al. (2024). Here we describe the experimental setup for fine-tuning these models.

- **CLIP-ViT-B/32 Models**: The CLIP-ViT-B/32 models are fine-tuned by Tang et al. (2024a). The Adam optimizer is employed with a fixed learning rate of $1e^{-5}$ for a total of 4,000 training steps with the batch size of 32. The zero-shot classification layer is computed on-the-fly with a frozen text encoder.

- **CLIP-ViT-L/14 Models**: Different from CLIP-ViT-B/32 models, these models are fine-tuned by Wang et al. (2024) with the training procedure described in Ilharco et al. (2022). The AdamW optimizer is employed with a fixed learning rate of $1e^{-5}$ for a total of 2,000 training steps with the batch size of 128, and a cosine annealing learning rate schedule with 200 warm-up steps. The zero-shot classification heads are pre-computed and frozen during fine-tuning process, following Ilharco et al. (2022) and Ortiz-Jimenez et al. (2024).

- **GPT-2 Models**: These models are fine-tuned by Tang et al. (2024a) with a constant learning rate of $5e^{-5}$ for 3 epochs.

- **Flan-T5-base and LoRA Models**: These models come from Tang et al. (2024a), with unspecified fine-tuning settings.

**Evaluation Metrics.** We measure performance using average task accuracy and normalized accuracy (relative to Fine-tuned baseline). For STSB (Wang, 2018), we use Spearman's correlation. Memory efficiency is evaluated by estimated memory footprint in Gb and normalized footprint (relative to Pre-trained baseline).

**Default Experimental Setup.** We use global random seeds 42, 43, 44 for three runs per experiment on random approaches. Each model's specific seed is generated by adding its index to the global seed, and is used consistently for layer shuffling and binary diagonal matrices across target layers. Following Ilharco et al. (2022), we apply uniform merging coefficients across models, optimized via grid search on validation sets (10% of training data, max 1,000 samples (Wang et al., 2024)). The search space is $0.1, 0.2, \cdots, 1.0$, extended to $0.1, 0.2, \cdots, 2.0$ for Flan-T5-base LoRA experiments.

## C  DERIVATION OF EQUATION 5

Here we derive the squared Frobenius norm of the interference $\lambda \sum_{i \neq k} \boldsymbol{T}_i^k$ in more details:

$$\left\| \lambda \sum_{i \neq k} \boldsymbol{T}_i^k \right\|_F^2 = \left\langle \lambda \sum_{i \neq k} \boldsymbol{T}_i^k, \lambda \sum_{i \neq k} \boldsymbol{T}_i^k \right\rangle_F , \tag{13}$$

$$= \lambda^2 \left( \sum_{i \neq k} \sum_{j \neq k} \langle \boldsymbol{T}_i^k, \boldsymbol{T}_j^k \rangle_F \right) , \tag{14}$$

$$= \lambda^2 \left( \sum_{i \neq k} \langle \boldsymbol{T}_i^k, \boldsymbol{T}_i^k \rangle_F + \sum_{i,j \neq k} \langle \boldsymbol{T}_i, \boldsymbol{T}_j^k \rangle_F \right) , \tag{15}$$

$$= \lambda^2 \left( \sum_{i \neq k} \|\boldsymbol{T}_i^k\|_F^2 + 2 \sum_{\substack{1 \leq i < j \leq n \\ i,j \neq k}} \langle \boldsymbol{T}_i^k, \boldsymbol{T}_j^k \rangle_F \right) , \tag{16}$$

$$= \lambda^2 \left( \sum_{i \neq k} \|\boldsymbol{T}_i^k\|_F^2 + 2 \sum_{\substack{1 \leq i < j \leq n \\ i,j \neq k}} \|\boldsymbol{T}_i^k\|_F \|\boldsymbol{T}_j^k\|_F \cos(\boldsymbol{T}_i^k, \boldsymbol{T}_j^k) \right) . \tag{17}$$

## D  ADDITIONAL ANALYSIS

### D.1  GPT-2 TEXT CLASSIFICATION EXPERIMENTS

We evaluate our proposed methods against established baselines by merging seven independently trained GPT-2 models on text classification tasks. As shown in Table 6, both our *STA+Shuffle* algorithm and its *TA+Shuffle* variants achieved significantly higher classification accuracy while doubling the memory footprint, consistent with the high performance on other benchmarks.

**Table 6:** Performance and memory comparison of GPT-2 models across seven GLUE text classification tasks, showing absolute and normalized accuracy (%), as well as memory footprint (Gb). Results averaged over three runs where applicable. Variances smaller than 0.1% are omitted.

| Method | Avg.(%) ↑ | Bits(Gb) ↓ | CoLA | MNLI | MRPC | QNLI | QQP | RTE | SST-2 |
|---|---|---|---|---|---|---|---|---|---|
| Pre-trained | 44.5 (54.3) | 0.498 (1.00) | 30.9 | 33.0 | 31.4 | 49.2 | 63.2 | 52.7 | 50.9 |
| Weight Averaging | 56.1 (63.3) | 0.498 (1.00) | 55.0 | 55.1 | 51.0 | 57.6 | 76.7 | 44.8 | 52.5 |
| Fisher Merging | 58.7 (64.7) | 0.498 (1.00) | 54.8 | 58.0 | 39.5 | 63.3 | 81.5 | 49.1 | 64.7 |
| RegMean | 68.8 (79.7) | 0.498 (1.00) | 61.7 | 70.4 | 65.4 | 69.7 | 78.8 | 56.0 | 79.7 |
| Task Arithmetic | 70.0 (85.4) | 0.498 (1.00) | 68.7 | 68.6 | 69.6 | 70.5 | 81.8 | 47.3 | 83.6 |
| Ties-Merging | 70.0 (82.4) | 0.498 (1.00) | 68.4 | 71.4 | 68.4 | 69.6 | 82.4 | 47.7 | 81.8 |
| PSP | 44.5 (54.3) | 0.498 (1.00) | 30.9 | 33.6 | 31.6 | 49.5 | 63.2 | 52.5 | 50.3 |
| Fine-tuned | 82.0 (100) | 3.49 (7.00) | 76.8 | 82.1 | 80.4 | 88.3 | 89.6 | 65.3 | 91.2 |
| **TA+Shuffle (Ours)** | **76.7** (93.5) | 0.997 (2.00) | **71.6** | 80.3 | **73.9** | 85.8 | 88.5 | 47.5 | 89.3 |
| **STA (Ours)** | 71.3 ±0.6 (87.0) | 0.997 (2.00) | 62.3 ±0.3 | 78.2 | 46.1 ±4 | 82.6 | 88.4 | **52.7** | 88.9 |
| **STA+Shuffle (Ours)** | 76.6 ±0.2 (93.4) | 0.997 (2.00) | 70.3 ±0.1 | **81.0** | 61.0 ±1.3 | **87.2** | **89.3** | **57.5 ±0.3** | **90.2** |

### D.2  DYNAMICS BETWEEN LAYER SHUFFLING AND SUPERPOSITION

We investigate how varying layer shuffling and superposition parameters affects model performance. We test target layer skip rates of $1, 2, 3, 4$, where only every k-th target layer within repetitive layer sets is shuffled and superposed. We also introduce *layer shifting* – a deterministic alternative to shuffling that shifts layers one position deeper with wrap-around – to study how

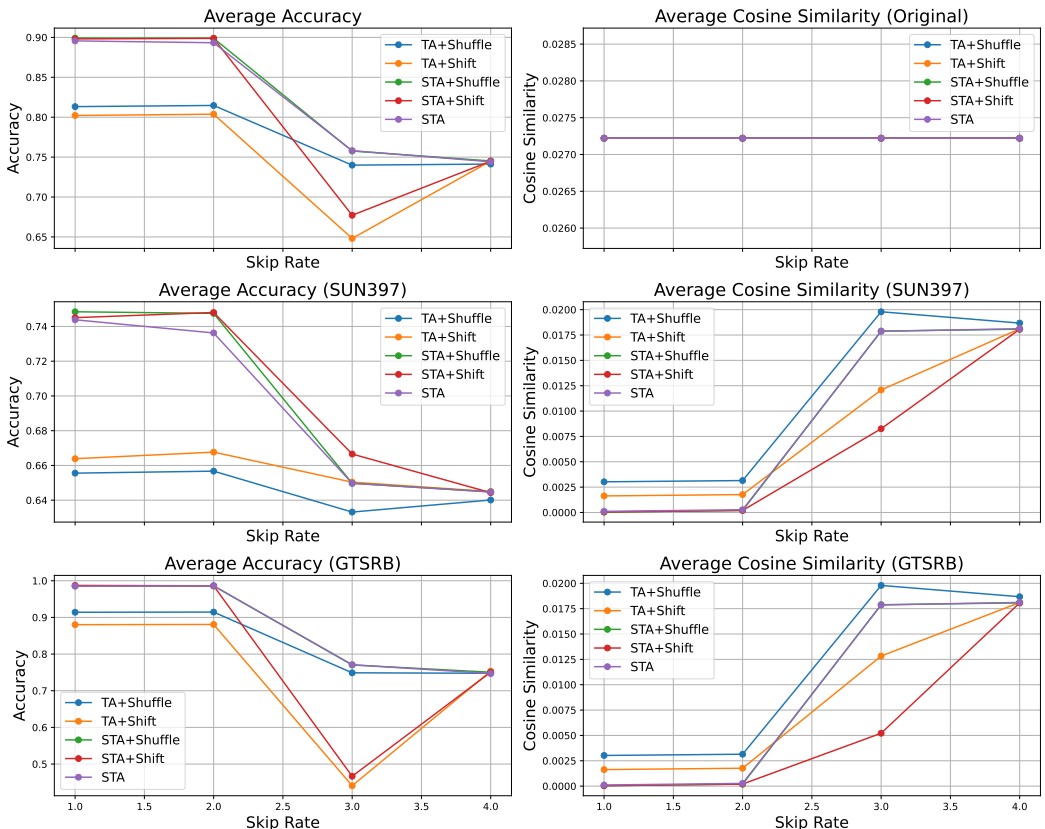

**Figure 6:** Average accuracy and cosine similarity among interfering task vectors when retrieving SUN397 and GTSRB models from 8 merged CLIP-ViT-B/32 models with various target layer skipping rates and shuffling/superposition setups.

different decorrelation approaches affect performance.

Experiments on eight CLIP-ViT-B/32 benchmarks (Figure 6) show averaged results across three repetitions, focusing on overall benchmark accuracy and two specific tasks: SUN397 (Xiao et al., 2010) and GTSRB (Stallkamp et al., 2012). We analyze both task performance and average pairwise cosine similarity among task vectors - both original and among interfering vectors during model retrieval.

As skip rate increases, accuracy declines while cosine similarity rises. Performance remains stable up to skip rate 2, suggesting potential memory savings through selective layer manipulation. For GTSRB, *TA+Shuffle* outperforms *TA+Shift* despite higher cosine similarity, indicating the method of achieving orthogonality matters beyond decorrelation levels. This pattern reverses for SUN397, revealing task-dependent variations and opportunities for task-specific optimization.

The correlation between interfering task vectors' cosine similarity and accuracy shows a negative trend (Figure 7), most pronounced in EuroSAT (Helber et al., 2019) and MNIST (Deng, 2012). While patterns vary across tasks, peak accuracy consistently occurs near zero cosine similarity. This observation, combined with *STA+Shift*'s strong performance at low skip rates (Figure 6), suggests a cosine similarity threshold may exist above which method selection and task properties become less critical.

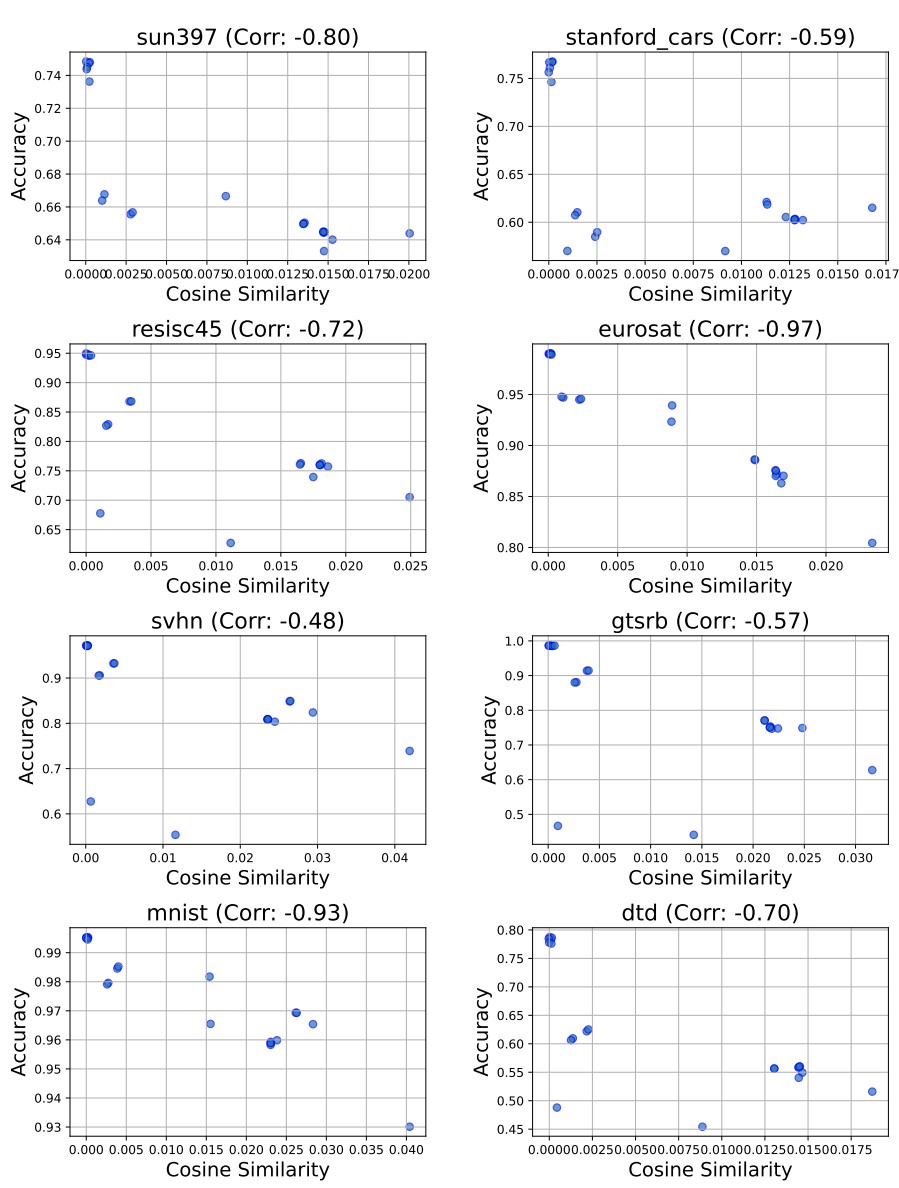

**Figure 7:** Correlation between the pairwise cosine similarity among interfering task vectors and the accuracy on the eight image classification tasks, with CLIP-ViT-B/32 merged with different levels of shuffling and superposition.

