# OpenReview forum: "Interfering with Interference: Blind Shuffling and Superposition for Better Multi-Model Compression"
_ICLR.cc/2025/Conference — Submitted to ICLR 2025_

### Official Review · Reviewer_jGUu · 2024-10-31

**Soundness:** 2
**Presentation:** 2
**Contribution:** 2
**Rating:** 5
**Confidence:** 3

**Summary:**

This paper introduces Layer Shuffling and Task Vector Superposition, two random mechanisms to reduce interference in multi-model compression by increasing orthogonality between task vectors. The methods achieve near-identical accuracy to individual fine-tuned models while reducing storage costs by 4 times and enable seamless hot-swapping of models without recomputation.

**Strengths:**

- The paper presents a novel approach to multi-model compression through random mechanisms

- The empirical evaluation is conducted across multiple benchmarks to demonstrate the effectiveness of the method

**Weaknesses:**

- The current presentation and writing require significant improvements. For instance, the mathematical analysis is overly simplistic and does not warrant extensive explanation. Additionally, the proposed method lacks a rigorous proof demonstrating why Layer Shuffling specifically enhances orthogonality more effectively than other potential random transformations.

- The interaction between Layer Shuffling and Task Vector Superposition isn't thoroughly analyzed as it's unclear whether they're truly complementary or if one method dominates the benefits

- The experiments are not convincing because the models used for comparison are generally much smaller, leading to expected inferior performance from competitors. Meanwhile, the authors' model is significantly larger, resulting in better performance, which does not necessarily demonstrate an advantage.

**Questions:**

see the above weakness part

---

> ### Author Response · Authors · 2024-11-28
>
> **1. Improving Presentation and Proof for Layer Shuffling Optimality**
>
> We thank the reviewer for their thoughtful feedback on the existing math analysis and we have removed redundant proofs from the appendix.
>
> Regarding the theoretical justification for Layer Shuffling's effectiveness: We recognize the challenge in providing a rigorous proof, as the method's performance inherently depends on learned parameter matrices, which is a product of the learning process. However, our empirical results, particularly in Figure 3(b) of the revision file, demonstrate that intra-model layer variation significantly exceeds inter-model variation for corresponding layers. This observation provides strong empirical support for Layer Shuffling's ability to enhance task vector orthogonality.
>
> We developed this heuristic and used it to establish layer shuffling as a simple and effective algorithm. While we acknowledge there may be better random transformations for task vector decorrelation, and it may be possible to design training procedures to enhance layer shuffling's effectiveness, we look forward to exploring these interesting directions in future work.
>
> **2. The Dynamics Between Layer Shuffling and Task Vector Superposition**
>
> Thank you for your insightful observation. To have a better look at the interaction between layer shuffling and task vector superposition, we curated below the average accuracy of all three variants of our method across all our benchmarks:
>
> |                          | **8xViT-B/32** | **8xFlan-T5-base** | **8xFlan-T5-base-LoRA** | **8xViT-L/14** | **14xViT-L/14** | **20xViT-L/14** | **7xGPT-2**  |
> | ------------------------ | -------------- | ------------------ | ----------------------- | -------------- | --------------- | --------------- | ------------ |
> | **Method**               | Acc.↑          | Acc.↑              | Acc.↑                   | Acc.↑          | Acc.↑           | Acc.↑           | Acc.↑        |
> | Pre-trained              | 48.2           | 75.7               | 75.7                    | 64.5           | 68.1            | 65.2            | 44.5         |
> | Fine-tuned               | 90.3           | 86.4               | 84.6                    | 94.4           | 93.5            | 94.2            | 82.0         |
> | **TA+Shuffle (Ours)**    | 81.3           | 85.7               | **83.9**                | 93.0           | 88.8            | 87.1±0.1        | ***76.7***   |
> | **STA (Ours)**           | **89.6**       | ***86.5***         | 83.0                    | **94.2**       | **92.8**        | **93.4**        | 71.3±0.6     |
> | **STA + Shuffle (Ours)** | ***89.9***     | **86.4**           | ***84.4***              | ***94.3***     | ***93.0***      | ***93.5***      | **76.6±0.2** |
>
> This reveals three key patterns in the interaction between these two mechanisms:
> 1. **Complementary Enhancement**: On several benchmarks (8xViT-B/32, 8xFlan-T5-base-LoRA, and 14xViT-L/14), we observe a clear complementary effect. The combined approach (STA + Shuffle) achieves notably higher accuracy than either method alone, demonstrating that the two techniques can work synergistically.
> 2. **Method Dominance**: In other cases, we find that one method may be primarily responsible for the performance gains.
> 3. **The Combined Approach Always Works**: The combined approach (STA + Shuffle) demonstrates remarkable consistency, achieving either the best or statistically indistinguishable from the best performance across all seven benchmarks.
>
> This interplay between the methods suggests a more nuanced relationship than simple additivity. Combining these two approaches does not necessarily lead to an additive effect. But when they are not additive, they are still complementary in that they seem to be more effective for complementary domains. Sometimes one dominates, sometimes the other. And the union always works.
>
> We have a working theory for this phenomenon. Our theoretical analysis (see equation 5 and section 4) shows that both techniques fundamentally work by enhancing orthogonality between task vectors, thereby reducing interference. The "one-side-dominant" benefits we observed in some cases may be due to diminishing returns in orthogonality enhancement: once sufficient orthogonality is achieved through either method, additional increases may yield minimal improvements. We plan to expand this analysis in a future version of our manuscript to provide a more detailed theoretical framework for understanding these interactions.

---

> > ### Author Response · Authors · 2024-11-28
> >
> > **3. Justifying Our Method on the Performance and Memory Tradeoff**
> >
> > We appreciate the reviewer’s insightful feedback regarding the comparison of model sizes and their impact on performance. To clarify the tradeoff between performance and memory usage, we present the average accuracy and memory footprint (in GB) of our method compared to baselines across multiple benchmarks:
> >
> >
> > |                          | **8xViT-L/14**        | **14xViT-L/14**       | **20xViT-L/14**       | **8xViT-B/32**        | **8xFlan-T5-base LoRA** |
> > | ------------------------ | --------------------- | --------------------- | --------------------- | --------------------- | ----------------------- |
> > | **Method**               | Acc.↑ (Bits↓)         | Acc.↑ (Bits↓)         | Acc.↑ (Bits↓)         | Acc.↑ (Bits↓)         | Acc.↑ (Bits↓)           |
> > | Pre-trained              | 64.5 (1.59)           | 68.1 (1.59)           | 65.2 (1.59)           | 48.2 (0.56)           | 75.7 (1.19)             |
> > | Fine-tuned               | 94.4 (10.53)          | 93.5 (18.18)          | 94.2 (25.84)          | 90.3 (2.84)           | 84.6 (1.25)             |
> > | Task Arithmetic          | 84.0 (***1.59***)     | 79.1 (***1.59***)     | 73.8 (***1.59***)     | 69.8 (***0.56***)     | 77.4 (***1.19***)       |
> > | TALL Mask + TA           | **94.2** (5.42)       | **92.4** (7.34)       | **93.2** (9.25)       | -                     | -                       |
> > | WEMoE                    | -                     | -                     | -                     | 89.2 (2.27)           | -                       |
> > | SMILE                    | -                     | -                     | -                     | **89.3** (1.23)       | 84.0 (1.21)             |
> > | **STA + Shuffle (Ours)** | ***94.3*** (**2.87**) | ***93.0*** (**2.87**) | ***93.5*** (**2.87**) | ***89.9*** (**0.89**) | ***84.4*** (**1.20**)   |
> >
> > Key observations:
> > 1. **SoTA Models Use More Memory But Perform Worse**: Compared to SoTA models (TALL-masks [1], WEMoE [4], SMILE [3]), our STA+Shuffle algorithm uses less memory while having higher accuracy.
> > 2. **Lightweight Baselines Underperform Despite Lower Memory Usage**: Although Task Arithmetic [2] uses only 55% the memory of our method, they exhibit substantially lower accuracy, especially as the number of tasks increases (e.g., 73.8% on 20xViT-L/14) or when merging LoRA models (e.g., 77.4% on 8xFlan-T5-base LoRA). These two scenarios are highly relevant to real-world applications.
> > 3. **Scalability in Real-World Scenarios**: As discussed in **[GR1]**, our method maintains high performance with a constant memory footprint even when more models are merged. This scalability distinguishes our approach from other baselines, making it more suitable for real-world applications where merging dozens of models may be necessary.
> >
> > In summary, our STA + Shuffle method demonstrates a favorable balance between performance and memory usage, outperforming both larger SoTA models and smaller baselines. This establishes a clear advantage in scenarios requiring efficient memory management without compromising accuracy.
> >
> >
> > [1] Wang, K., Dimitriadis, N., Ortiz-Jimenez, G., Fleuret, F., & Frossard, P. (2024). Localizing Task Information for Improved Model Merging and Compression. arXiv preprint arXiv:2405.07813.
> >
> > [2] Ilharco, G., Ribeiro, M. T., Wortsman, M., Gururangan, S., Schmidt, L., Hajishirzi, H., & Farhadi, A. (2022). Editing models with task arithmetic. arXiv preprint arXiv:2212.04089.
> >
> > [3] Tang, A., Shen, L., Luo, Y., Xie, S., Hu, H., Zhang, L., ... & Tao, D. (2024). Smile: Zero-shot sparse mixture of low-rank experts construction from pre-trained foundation models. arXiv preprint arXiv:2408.10174.
> >
> > [4] Tang, A., Shen, L., Luo, Y., Yin, N., Zhang, L., & Tao, D. (2024). Merging Multi-Task Models via Weight-Ensembling Mixture of Experts. arXiv preprint arXiv:2402.00433.

---

> > ### Author Response · Authors · 2024-11-28
> > **A Brief Follow-up**
> >
> > We have also added a detailed analysis of the interaction between layer shuffling and superposition in **[GR3]** and **Appendix D.2** in the revision file, which provides further empirical insights into their dynamics and task-dependent effectiveness. Thank you again for helping us improve this important aspect of our work.

---

### Official Review · Reviewer_mt5T · 2024-11-02

**Soundness:** 3
**Presentation:** 3
**Contribution:** 3
**Rating:** 6
**Confidence:** 4

**Summary:**

The paper introduces two methods, Layer Shuffling and Task Vector Superposition, aimed at reducing interference when compressing multiple fine-tuned models into a single multitask model using task vector arithmetic. Layer Shuffling works by randomly reordering layers in each model before merging, reducing alignment between task vectors. Task Vector Superposition applies random orthogonal transformations to further decorrelate task vectors. Both techniques minimize interference and improve performance across tasks. Experiments with CLIP-ViT, Flan-T5, and GPT-2 show that this approach achieves higher accuracy than vanilla task arithmetic and other baseline methods.

**Strengths:**

- The paper makes an observation that individual task vectors are too similar and successfully uses it to reduce task vector interference, leading to better multitask performance in model compression scenarios.
- Both proposed techniques operate without needing data, allowing flexible model addition or removal without retraining or optimization.
- The method achieves storage reduction compared to keeping individual models.
- The approach is shown to improve performance across diverse domains including image classification, text generation, and text classification.
- The method enables on-the-fly model integration, allowing seamless "hot-swapping" of models.
- The paper is very well written and clearly structured.

**Weaknesses:**

- While the paper compares its method to several baseline techniques, it misses comparison with closely related recent works, particularly Guillermo Ortiz-Jimenez et al.'s work (mentioned in the paper) on task vector manipulation for model merging. Including these comparisons would strengthen the submission.
- Although the authors claim minimal memory overhead, additional context matrices and shuffled task vectors nearly double the memory requirement, which may not always justify the marginal performance gains over baselines like SMILE.
- LoRA results show that SMILE achieves a better tradeoff between accuracy and memory than the reported combination of the proposed methods.

**Questions:**

Q1: Although randomization offers clear advantages like data independence, would a more systematic approach to orthogonalizing task vectors further improve the performance?
Q2: Did you observe any (in-)consistent performance variance due to randomness in shuffling and superposition?

---

> ### Author Response · Authors · 2024-11-28
>
> **1. Including Comparisons with Recent Task Vector Manipulation Methods**
>
> We appreciate the suggestion to compare our algorithm with recent task vector manipulation methods, specifically TALL-masks [1]. We have now evaluated our approach against TALL-masks on merging 8, 14, and 20 ViT-L/14 models. The experiment results and discussion are now included in **[GR2]** and the “Scalability Analysis” section of our revision.
>
> **2. Clarifying Memory Overhead and Performance Tradeoff with SMILE as a Comparison**
>
> Thank you for raising this important consideration about memory efficiency. As discussed in the "Amortizable Memory Overhead" section of our paper and in **[GR1]**, our method requires only storing one additional task vector instance beyond the pre-trained model. Subsequent models can be merged by saving just their random seeds. This results in a constant memory footprint, which becomes increasingly advantageous when merging more and larger models, as elaborated in **[GR2]**.
>
> To address your concern about whether our memory overhead justifies the performance gains over baselines like SMILE [2], we have prepared a comparison between our method and SMILE across multiple benchmarks. The average accuracy and storage cost (in GB) are presented below:
>
> **Performance and Memory Footprint Comparison with SMILE Across Benchmarks**
> |                          | **8xViT-B/32**      | **8xFlan-T5-base** | **8xFlan-T5-base LoRA** |
> | ------------------------ | ------------------- | ------------------ | ----------------------- |
> | **Method**               | Acc.↑ (Bits↓)       | Acc.↑ (Bits↓)      | Acc.↑ (Bits↓)           |
> | Pre-trained              | 48.2 (0.56)         | 75.7 (1.19)        | 75.7 (1.19)             |
> | Fine-tuned               | 90.3 (2.84)         | 86.4 (9.52)        | 84.6 (1.25)             |
> | SMILE                    | 89.3 (1.23)         | 85.5 (**1.81**)    | 84.0 (1.21)             |
> | **STA + Shuffle (Ours)** | **89.9** (**0.89**) | **86.4** (2.38)    | **84.4** (**1.20**)     |
>
> Our STA+Shuffle algorithm outperforms SMILE across all benchmarks, while using less memory on the 8xViT-B/32 and 8xFlan-T5-base LoRA tasks. Although we use more memory than SMILE on the 8xFlan-T5-base benchmark, our performance matches that of the fine-tuned baseline, effectively saturating this benchmark. We plan to compare our method with SMILE on more challenging scenarios (e.g., merging 20 ViT-L/14 models) and will include those results in the next version of our manuscript.
>
> **3. Comparing Accuracy and Memory Trade-offs with SMILE on LoRA Models**
>
> Thank you for raising this point. We discovered that in our initial experiment, we had not properly tuned the merging coefficient lambda, instead using a fixed value of 1.0. This was suboptimal compared to coefficients used in other benchmarks. After correcting this oversight, we performed a grid search on the validation set to find the optimal lambda value. This adjustment led to significantly improved results, as shown in the following table:
>
> **Multi-task performance when merging Flan-T5-base LoRA models on eight GLUE tasks**
> | **Method**             | **Avg.(%) ↑**     | **Bits(Gb) ↓**    |
> | ---------------------- | ----------------- | ----------------- |
> | Pre-trained            | 75.7 (87.6)       | 1.19 (1.00)       |
> | Fine-tuned             | 84.6 (100)        | 1.25 (1.05)       |
> | Task Arithmetic        | 77.4 (91.5)       | ***1.19 (1.00)*** |
> | SMILE                  | **84.0 (99.3)**   | 1.21 (1.02)       |
> | **STA+Shuffle (Ours)** | ***84.4 (99.8)*** | **1.20 (1.01)**   |
>
> The updated results demonstrate that our STA+Shuffle algorithm surpasses SMILE [2], achieving higher accuracy while maintaining lower memory requirements. We have incorporated this revised analysis into the manuscript’s “Parameter Efficient Finetuning (PEFT) Model Compression” section.

---

> > ### Author Response · Authors · 2024-11-28
> >
> > **Q1: Although randomization offers clear advantages like data independence, would a more systematic approach to orthogonalizing task vectors further improve the performance?**
> >
> > We agree that systematically orthogonalizing task vectors could further enhance performance. As shown in Figure 2 and Table 1, there is a clear correlation between task vector orthogonality and performance, and current vectors are not perfectly perpendicular after shuffling and superposition. An interesting direction for future work would be to maximize orthogonality based on model parameters and analyze the resulting performance changes. However, such a parameter-dependent approach may make it difficult to specify task-specific operations in a memory-efficient way or enable incremental model addition and subtraction. Therefore, we focused on our random algorithm, which effectively addresses these challenges and is well-suited for resource-constrained environments.
> >
> > **Q2: Did you observe any (in-)consistent performance variance due to randomness in shuffling and superposition?**
> >
> > Our analysis of accuracy variance across all variants showed minimal variation, with values below 0.1% (not shown in tables) in most cases. The only exceptions were certain tasks using GPT-2.
> >
> > [1] Wang, K., Dimitriadis, N., Ortiz-Jimenez, G., Fleuret, F., & Frossard, P. (2024). Localizing Task Information for Improved Model Merging and Compression. arXiv preprint arXiv:2405.07813.
> >
> > [2] Tang, A., Shen, L., Luo, Y., Xie, S., Hu, H., Zhang, L., ... & Tao, D. (2024). Smile: Zero-shot sparse mixture of low-rank experts construction from pre-trained foundation models. arXiv preprint arXiv:2408.10174.

---

### Official Review · Reviewer_JsQG · 2024-11-03

**Soundness:** 2
**Presentation:** 2
**Contribution:** 3
**Rating:** 3
**Confidence:** 5

**Summary:**

The paper proposes two stochastic mechanisms to improve performance for multi-task model merging by reducing task interference. First, the method takes advantage of the repeating structure of modern neural networks and randomly shuffles the same-module layer across blocks by first showing that the layers are mostly similar in the within-block across tasks. Second, the paper proposes random binary matrices to multiply parameter vectors to further reduce the task vector similarity. During inference, the inverse transforms are applied. The paper performs experiments across diverse benchmarks.

**Strengths:**

1. The method is intuitive and simple. The motivation for both components is well written and properly ablated.
2. The method is scalable and memory efficient (modulo the duplication of model parameters) given that it only requires the storage of random seeds to retrieve the final model.
3. The experimental results are strong across benchmarks.

**Weaknesses:**

1. The method has a limitation that is not discussed a lot, apart from the title: it requires the knowledge of the task id during inference. This needs to be underlined during the comparison with methods such as ties and task arithmetic for fairness.
2. lack of forward pass time complexity comparison. The proposed method introduces an overhead in the forward pass: layers need to be reshuffled in the correct order, signs need to be restored and the residual needs to be added to the pre-trained weights. Therefore, there should be a study of how much overhead all these operations incur.
3. Missing baselines: Given the parameter increase and the time complexity overhead, the paper should compare with the compression algorithm of [1].
4. The paper solely focuses on small models, base variants on ViT and Flan-T5, but the literature uses ViT-L/14 and T5-XXL regularly. It would also be interesting to check the performance of the method as tasks increase, see 14 and 20 task benchmarks from [1]. It would be interesting to also track the forward pass time metrics in the case of larger models.
5. L268-269: fix references for benchmarks: the vision one for instance comes from Ilharco et al. and not from FusionBench
6. Baselines and their categorization are not explained and the reader cannot understand why PSP ins included given its poor results or what WEMoE and SMILE are on their own category compared to everything else. It would be helpful for the reader to provide a brief description of each method as well as a high level overview of the categories to help the reader understand rather than deferring to the appendix where they are actually not discussed.
7. Extremely limited Related work: the quality of the paper is heavily undermined by the lack of proper references and discussion over related work.

Minor Comments


- L202: ontain → obtain
- Rephrase informal writing:
	- L217: “which we expect to be much lower”
	- L150: “but this balance has generally been tricky to achieve”

[1] Wang, K., Dimitriadis, N., Ortiz-Jimenez, G., Fleuret, F. and Frossard, P., 2024. Localizing Task Information for Improved Model Merging and Compression. *arXiv preprint arXiv:2405.07813*.

**Questions:**

1. why is 5.03 the number of Gb attributed to fine-tuned? shouldn’t it be 8x the pre-trained model?

---

> ### Author Response · Authors · 2024-11-28
>
> **1. Forward Pass Time Complexity Comparison**
>
> Thank you for this important question. As detailed in **[GR1]**, we evaluated our method's runtime overhead when using random seeds to retrieve models with shuffling and superposition applied to all layers. The retrieval process on an Intel Xeon Gold 6448Y CPU averaged 292.70 ms for CLIP-ViT-B/32 and 658.19 ms for CLIP-ViT-L/14 over 10 runs. These overheads remain constant regardless of the number of merged models, as our method introduces no additional memory requirements. GPU optimization could further reduce these runtimes. These measurements have been added to the "Amortizable Memory Overhead" section in our revision, and an asymptotic complexity analysis will be included in the next version of the paper.
>
> **2. Missing Baselines from TALL-masks**
>
> Thank you for pointing this out. As noted in **[GR2]**, we've added comparisons with TALL-masks [1] algorithms, specifically TALL-masks + TA and Magnitude Masking. We excluded Magnitude Pruning due to its consistently lower performance and missing implementation. While TALL-masks can be enhanced with orthogonal techniques like TIES [2], we excluded this variant for fair comparison. In future version of our paper, we plan to enhance our approach with techniques like Adamerging [3] and include comparisons with TALL-masks+TIES.
>
> **3. Extending Experiments to Larger Models and More Tasks**
>
> Thank you for your suggestion. We have conducted new experiments merging 8, 14, and 20 ViT-L/14 models as discussed in [GR2]. We will fine-tune and merge even larger models in greater quantities, and include the results in our paper.
>
>
> **4. Discussing Task ID Requirement During Inference for Fair Comparisons**
>
> We acknowledge that our method requires the task ID during inference and will emphasize this in our comparison with other methods. However, we believe the comparison remains fair for the following reasons:
> 1. **Other Baselines Also Require Task IDs**: For example, when merging image classification models, all baselines require a task ID to select the task-specific classification head for inference. This requirement is made clear in Fisher Merging [7] but not explicitly discussed in many other baselines.
> 2. **Task IDs Enable Strong Performance**: Leveraging task IDs allows task-specific retrieval, contributing to our method’s high performance and low memory footprint. This is a good feature and aligns with the goals of model merging and compression.
> 3. **Task Identification is Context-Dependent**: In completely task-agnostic environments, it’s possible to infer task ID from the input like WEMoE [5] and SMILE [6]. But in many real-world scenarios where task-specific queries or datasets are naturally tagged with task IDs (e.g., multi-tenant APIs or application-specific deployments), task IDs are easy to come by. The system-specific nature of task identification and its orthogonality to our core contributions have led us to focus on other aspects rather than developing a separate task identification mechanism.
>
> To address your concern comprehensively, we will incorporate a task classifier to our method in the next version of our paper.
>
> **5. Providing Detailed Explanations and Categorization of Baseline Methods**
>
> Thank you for these thoughtful suggestions. We have enhanced the clarity of our work by:
> * Revising the Experimental Setup section to better explain model categorization
> * Adding Appendix B with comprehensive details on all baselines, datasets, models, and experimental setups
> * Clarifying PSP [8]'s role as an online learning model superposition baseline adapted to demonstrate the effectiveness of our task vector superposition approach in the offline setting
>
>
> **6. Expanding and Enhancing the Related Work Section**
>
> We thank the reviewer for this important feedback. We have thoroughly expanded our Related Work section to provide a more comprehensive coverage of model merging and compression literature. The revised manuscript now includes detailed discussions and proper references in these areas. We welcome specific suggestions for any missing critical references to include in the final version.
>
> **7. Question: Why is 5.03 the number of Gb attributed to fine-tuned? shouldn’t it be 8x the pre-trained model?**
>
> Thank you for this important question. The 5.03x figure for fine-tuned models appears lower than 8x the pre-trained model size because our CLIP models from FusionBench [4] use a frozen text encoder during fine-tuning. Only the vision encoders need to be stored separately for individual fine-tuned models. We have clarified this in the manuscript.

---

> > ### Author Response · Authors · 2024-11-28
> >
> > **8. Minor Fixes**
> >
> > * L268-269: fix references for benchmarks: the vision one for instance comes from Ilharco et al. and not from FusionBench.
> >   * We apologize for the mistake, and have fixed it in the revision.
> > * L202: ontain → obtain
> >   * We have fixed the typo. Thank you.
> > * Rephrase informal writing: L217: “which we expect to be much lower”
> >   * We have refined it to ““which is expected to be significantly lower due to the reduced alignment of parameter vectors from different layers.” Thank you.
> > * Rephrase informal writing: L150: “but this balance has generally been tricky to achieve”
> >   * We have modified it to “but attaining this balance has often been challenging”. Thank you.
> >
> >
> >
> > [1] Wang, K., Dimitriadis, N., Ortiz-Jimenez, G., Fleuret, F., & Frossard, P. (2024). Localizing Task Information for Improved Model Merging and Compression. arXiv preprint arXiv:2405.07813.
> >
> > [2] Yadav, P., Tam, D., Choshen, L., Raffel, C. A., & Bansal, M. (2024). Ties-merging: Resolving interference when merging models. Advances in Neural Information Processing Systems, 36.
> >
> > [3] Yang, E., Wang, Z., Shen, L., Liu, S., Guo, G., Wang, X., & Tao, D. (2023). Adamerging: Adaptive model merging for multi-task learning. arXiv preprint arXiv:2310.02575.
> >
> > [4] Tang, A., Shen, L., Luo, Y., Hu, H., Du, B., & Tao, D. (2024). Fusionbench: A comprehensive benchmark of deep model fusion. arXiv preprint arXiv:2406.03280.
> >
> > [5] Tang, A., Shen, L., Luo, Y., Xie, S., Hu, H., Zhang, L., ... & Tao, D. (2024). Smile: Zero-shot sparse mixture of low-rank experts construction from pre-trained foundation models. arXiv preprint arXiv:2408.10174.
> >
> > [6] Tang, A., Shen, L., Luo, Y., Yin, N., Zhang, L., & Tao, D. (2024). Merging Multi-Task Models via Weight-Ensembling Mixture of Experts. arXiv preprint arXiv:2402.00433.
> >
> > [7] Matena, M. S., & Raffel, C. A. (2022). Merging models with fisher-weighted averaging. Advances in Neural Information Processing Systems, 35, 17703-17716.
> >
> > [8] Cheung, B., Terekhov, A., Chen, Y., Agrawal, P., & Olshausen, B. (2019). Superposition of many models into one. Advances in neural information processing systems, 32.

---

### Official Review · Reviewer_gJhr · 2024-11-04

**Soundness:** 3
**Presentation:** 2
**Contribution:** 3
**Rating:** 6
**Confidence:** 4

**Summary:**

This paper introduces two methods, layer shuffling and task vector superposition, aimed at reducing interference between task vectors in multi-model compression scenarios. The proposed methods work by increasing the orthogonality of task vectors, thus minimizing their interference during merging. By leveraging randomization, these methods require no additional training and can be applied across various models and tasks. Experiments on multiple benchmarks, including CLIP, Flan-T5, and GPT-2, demonstrate that this approach can achieve comparable performance to fine-tuned models while reducing storage costs, particularly for real-world deployment scenarios where adding or removing models on the fly is necessary.

**Strengths:**

* Simplicity and Effectiveness: One of the major strengths of this paper lies in its approach’s simplicity. Layer shuffling and task vector superposition are straightforward yet powerful techniques that effectively reduce interference without needing additional training, optimization, or complex configurations. This simplicity not only enhances the practicality of the approach but also makes it easy to implement and adapt across various multi-model compression tasks, proving that even minimal adjustments can yield significant performance improvements.

* Effective Interference Reduction: The combination of layer shuffling and task vector superposition is innovative in addressing interference by increasing orthogonality among task vectors. This approach allows for a more effective merging process, yielding improved model accuracy without the need for additional optimization or training steps.

* Adaptability and Scalability: The proposed method’s flexibility is a clear strength. Its data and model-independent nature enables seamless additions and removals of models (hot-swapping) without re-computation, a valuable feature for dynamic applications. Moreover, the approach is efficient, doubling the memory footprint while providing significant accuracy improvements.

* Comprehensive Evaluation: The experiments cover a range of benchmarks and tasks, showcasing the model’s capability across various domains, from image classification to text generation. This breadth of evaluation helps establish the generalizability of the method across tasks and model architectures.

**Weaknesses:**

* Lack of Detailed Performance Analysis Based on Shuffle/Superposition Levels: It would be useful to analyze the impact of different levels of shuffling and superposition, as these levels could influence task vector similarity and interference differently. This analysis would provide a clearer picture of optimal interference reduction strategies.
* Clarity Issues in Method Description: Some aspects of the method, such as the merged task vector formation in equation (8), could benefit from further clarification. Specifically, does shuffling task vectors in different layers cause mixing of task vectors across layers, for instance, between k-1 or k+1? Clarifying this would enhance understanding of how the shuffle affects layer-specific task vector alignment.
* Effectiveness Across Tasks: The effectiveness of either TA+Shuffle or STA appears to vary by task, yet the paper does not discuss why some tasks benefit more from specific strategies. A more in-depth analysis here would provide insights into optimizing methods based on task characteristics.
* Related Work Reference (PEFT): (Maybe, long shot) this paper is related? "Efficient Storage of Fine-Tuned Models via Low-Rank Approximation of Weight Residuals,"
* Minor Formatting Issues: There are some minor formatting errors in the document, such as incorrect Latex punctuation and inconsistent reference formatting. For example, equation 3 is mistakenly referenced in the context of equation 4, and parentheses are missing in certain citations. Additionally, clarifying what the values in parentheses mean in tables, such as in the average (%) and Bits (Gb) columns, would be helpful, as it currently requires reading the text to understand that they refer to relative performance to fine-tuned models.

**Questions:**

Please see the above.

---

> ### Author Response · Authors · 2024-11-28
>
> **1. Investigating the Impact of Shuffling and Superposition Levels on Performance**
>
> We appreciate this valuable suggestion. Following it, we conducted an extensive analysis of shuffling and superposition levels in the newly added **Appendix D.2**, which has been discussed in **[GR3]**. Our analysis revealed that performance remains robust at lower skip rates, suggesting opportunities for memory optimization. We discovered a critical cosine similarity threshold that helps explain the effectiveness of different interference reduction methods. We hope these findings provide a clearer picture about optimal interference reduction approaches. Thank you for helping us improve our work.
>
> **2. Analyzing Task-Specific Effectiveness of Proposed Strategies**
>
> Thank you for this insightful comment about task-specific effectiveness. As shown in **[GR3]**, we observed interesting task-dependent variations - for example, TA+Shift underperformed TA+Shuffle on GTSRB but showed reversed behavior on SUN397. While our current task set is limited for a comprehensive statistical analysis of task characteristics, we have included initial findings about these variations in **Appendix D.2**. We acknowledge this is an important direction and plan to conduct a more thorough analysis with a larger set of tasks in future work.
>
>
> **3. Clarifying Merged Task Vector Formation and Layer Shuffling Effects**
>
> We appreciate you bringing this to our attention. Yes indeed, layer shuffling can mix layer k-1 from model A with layer k+1 from model B, given that they have the same shape. We will enrich these text descriptions and incorporate them to a future version of our paper. Meanwhile, to help readers understand layer shuffling more easily, we have added a visual illustration in Figure 3 (a) of our revision file.
>
> **4. Incorporating Relevant PEFT Literature into Related Work**
>
> We appreciate the reviewer's suggestion regarding this relevant work. We have added a brief discussion of their paper in the updated Related Work section. Our approach shares similarities in that both methods aim to reduce the memory footprint of delta parameters from fine-tuned models. The fundamental difference is that their method focuses on compressing each individual model's residual separately, whereas we leverage the parameter properties of a set of aligned fine-tuned models to compress them collectively, thereby reducing cross-model redundancy.
>
> We would like to compare our method to theirs. However, we are currently unable to reproduce their results due to the inaccessibility of their codebase, as the provided link is broken. We have contacted the authors to request access to their code and will include a comparison with their methods in a future version of our paper if we are able to obtain it.
>
>
> **5. Minor Formatting Issues**
>
> Thank you for your helpful suggestions! We have addressed all the formatting issues, including correcting the LaTeX equation punctuation and ensuring consistent reference formatting throughout the document. Additionally, we have clarified the values in parentheses within the tables, such as the "average (%)" and "Bits (Gb)" columns, by providing detailed explanations in the table captions, ensuring that the meaning is clear without needing to refer back to the text.

---

### Author Response · Authors · 2024-11-28
**General Responses**

We sincerely thank all reviewers for their thoughtful feedback and suggestions. Below, we address the key concerns regarding memory-performance tradeoffs, model scaling, and the interplay between shuffling and superposition. The revisions and corresponding results have been incorporated into the manuscript, with all changes highlighted in blue.


**[GR1] Memory Efficiency Clarification (Reviewer JsQG, mt5T, and jGUu)**

We appreciate the reviewers' valid concerns regarding the memory footprint of our methods. To address these concerns, we present new findings demonstrating that our approach requires saving only the pre-trained model and a merged task vector. Merging additional models involves storing just a single 8-bit random seed per model. Our method introduces less than 1s overhead during the forward pass for retrieving each model and maintains high performance. Note this runtime overhead is not dependent on the number of merged models.

1. **Compressing Additional Models into 8-bit Random Seeds**

Our methods—random layer shuffling and task vector superposition—are random operations independent of both data and model parameters. This property allows us to retrieve each merged model by reconstructing the layer shuffling orders and the inverses of the context matrices from a single random seed (takes 8-bit if less than 256 models are merged). Although we double the memory footprint due to the need to manipulate the merged task vector differently for each task, this initial investment remains fixed, regardless of the number of models being merged.

2. **Achieving Near-Fine-Tuned Performance with Reduced Memory Usage**

Our methods achieve performance close to that of fine-tuned models across all testing scenarios. Thanks to the feedback from Reviewer JsQG, we realized that when merging CLIP-ViT-B/32 models, our methods consume less memory than previously estimated, as we mistakenly counted the text encoder twice. Here is the updated Table 1 (truncated to highlight key comparisons):

**Performance comparison on eight image classification tasks with CLIP-ViT-B/32 models**
| **Method**             | **Avg.(%) ↑**     | **Bits(Gb) ↓**     |
| ---------------------- | ----------------- | ------------------ |
| Pre-trained            | 48.2 (53.4)       | 0.564 (1.00)       |
| Fine-tuned             | 90.3 (100)        | 2.84 (5.03)        |
| Task Arithmetic        | 69.8 (77.2)       | ***0.564 (1.00)*** |
| WEMoE                  | 89.2 (98.8)       | 2.27 (4.03)        |
| SMILE                  | **89.3 (98.9)**       | 1.23 (2.20)        |
| **STA+Shuffle (Ours)** | ***89.9 (99.6)*** | **0.89 (1.58)**    |

This updated table demonstrates that our algorithm achieves near-perfect accuracy while using significantly less memory compared to SMILE [3] and WEMoE [4].

Moreover, as shown in **[GR2]**, our STA+Shuffle algorithm maintains accuracy close to that of individually fine-tuned baselines when merging 14 or even 20 CLIP-ViT-L/14 models, with a constant memory footprint of less than twice that of a single model. In contrast, the state-of-the-art baseline TALL-masks uses 323% more memory than our method while achieving slightly lower performance when merging 20 CLIP-ViT-L/14 models. Finally, **[GR3]** shows that we can **halve** the current memory footprint by selecting a subset of layers to shuffle and superpose.

While methods like Task Arithmetic consume approximately 50% less memory than ours, their performance is significantly lower, with accuracy scores resembling those of zero-shot pre-trained models when more and larger models are merged, as noted in **[GR2]**.

3. **Minimal Runtime Overhead in Forward Pass for Model Retrieval**

To evaluate the overhead introduced by our method, we conducted experiments using a single random seed to retrieve models with shuffling and superposition applied to all layers. For the CLIP-ViT-B/32 model, the retrieval process executed on an Intel Xeon Gold 6448Y CPU over 10 repetitions yielded an average runtime of 292.70 ms. For the larger CLIP-ViT-L/14 model, the average runtime was 658.19 ms under the same conditions. We anticipate that further optimizations, such as utilizing GPUs, could reduce these runtimes further. As our method shows no increase in memory footprint as more models are merged, these overhead measurements also apply to scenarios where 14 or 20 CLIP-ViT-L/14 models are merged.

In conclusion, the new findings show that our algorithm can achieve near-perfect accuracy with constant memory use and minimal runtime overhead. It outperforms state-of-the-art methods in performance and memory efficiency, offering a simple yet powerful solution for model merging and serving.

---

> ### Author Response · Authors · 2024-11-28
>
> **[GR2] Benchmarking Against TALL-masks on Larger Models and More Tasks (Reviewer JsQG and mt5T)**
>
> Reviewers JsQG and mt5T requested performance comparisons with recent works, particularly the compression baselines from TALL-masks [1]. Reviewer JsQG also emphasized the need for evaluation on a broader range of tasks (e.g., 14 and 20 tasks) and larger models (e.g., ViT-L/14, T5-XXL). In response, we present the performance of our method on the ViT-L/14 benchmark across 8, 14, and 20 tasks, as provided in TALL-masks [1].
>
> The average accuracy and storage cost (in GB) estimates are presented here. These scores differ slightly from those reported in TALL-masks [1] due to corrections we made to dataset split issues in SUN397, EuroSAT, and DTD. Furthermore, the storage cost figures vary from the previously reported values, as the original estimates were inaccurate.
>
> **ViT-L/14 Performance Comparison with 8, 14, and 20 Tasks**
> |                          | **8 tasks**           | **14 tasks**          | **20 tasks**          |
> | ------------------------ | --------------------- | --------------------- | --------------------- |
> | **Method**               | Acc.↑ (Bits↓)         | Acc.↑ (Bits↓)         | Acc.↑ (Bits↓)         |
> | Pre-trained              | 64.5 (1.59)           | 68.1 (1.59)           | 65.2 (1.59)           |
> | Task Arithmetic          | 84.0 (***1.59***)     | 79.1 (***1.59***)     | 73.8 (***1.59***)     |
> | Magnitude Masking        | 92.8 (5.42)           | 90.6 (7.34)           | 90.9 (9.25)           |
> | TALL-masks + TA           | **94.2** (5.42)       | **92.4** (7.34)       | **93.2** (9.25)       |
> | **STA + Shuffle (Ours)** | ***94.3*** (**2.87**) | ***93.0*** (**2.87**) | ***93.5*** (**2.87**) |
> | Fine-tuned               | 94.4 (10.53)           | 93.5 (18.18)          | 94.2 (25.84)          |
>
> Key observations:
> 1. **High Performance**: Our STA + Shuffle algorithm consistently outperforms alternative methods when merging ViT-L/14 models. Specifically, it achieves performance close to that of individually fine-tuned models, even when merging as many as 14 or 20 ViT-L/14 models.
> 2. **Efficient Storage**: While TALL-masks + TA [1] exhibits a gradual increase in storage costs as more models are merged, our STA + Shuffle algorithm maintains a stable storage cost of less than twice the size of a single model, regardless of the number of models merged. This efficiency is achieved because we only need to store a single random seed for each model. Models can then be retrieved with negligible latency—292.70 ms for a single ViT-B/32 and 658.19 ms for a single ViT-L/14—measured over 10 repetitions on an Intel Xeon Gold 6448Y CPU.
> 3. **Comparison to Lightweight Baselines**: Although model-merging baselines like Task Arithmetic [2] use 55% the storage of our method, their performance is inferior, resembling pre-trained performance levels as the number of tasks increases.
>
> We have updated our paper to include these comparisons (see **Section 5.9 Scalability Analysis**), emphasizing our algorithm’s efficiency and effectiveness in merging larger and more numerous models.

---

> ### Author Response · Authors · 2024-11-28
>
> **[GR3] Dynamics Between Layer Shuffling and Superposition (Reviewer gJhr and jGUu)**
>
> Reviewers gJhr and jGUu noted that the interaction between layer shuffling and superposition wasn't thoroughly analyzed, particularly their effectiveness variation across different tasks. Following these suggestions, we conducted a detailed analysis in the newly added **Appendix D.2**, which revealed several important insights:
> 1. **Higher Skip Rates Increase Interference**: We introduced "skip rates" where a skip rate of k means only every k-th target layer within repetitive layer sets is shuffled and superposed. Higher skip rates led to increased cosine similarity and decreased performance.
> 2. **Performance Remains Robust at Lower Skip Rates**: Performance maintains stability up to skip rate 2 (Figure 6), suggesting potential for halving memory footprint through selective layer manipulation.
> 3. **Decorrelation Method Impact**: We introduced layer shifting as a deterministic alternative to layer shuffling, where layers move one position deeper with wrap-around. Notably, for GTSRB, TA+Shuffle outperformed TA+Shift despite higher cosine similarity, indicating that the method of achieving orthogonality matters beyond the decorrelation level.
> 4. **Task-Dependent Variations**: While TA+Shift underperformed TA+Shuffle on GTSRB, this pattern reversed for SUN397. The correlation between cosine similarity and accuracy also varied across tasks (Figure 7), highlighting opportunities for task-specific optimization.
> 5. **Cosine Similarity Threshold Hypothesis**: Despite these method and task-specific behaviors, we found that optimal performance consistently occurs at the lowest cosine similarity (Figure 7). This led us to hypothesize a critical cosine similarity threshold below which method selection and task properties become less important. This explains the dominant effectiveness noted by Reviewer jGUu: the STA method achieves near-zero cosine similarity on its own, pushing below this threshold for strong performance. Adding layer shifting (STA+Shift) or shuffling (STA+Shuffle) only marginally reduces the already-low similarity, yielding minimal gains. In contrast, layer shifting/shuffling alone cannot reduce similarity enough to cross this threshold, resulting in inferior performance.
>
> We have highlighted these insights in **Appendix D.2** to help develop more effective interference reduction strategies. We plan to explore more scenarios and expand our cosine similarity threshold hypothesis in the next version of our paper.
>
>
> [1] Wang, K., Dimitriadis, N., Ortiz-Jimenez, G., Fleuret, F., & Frossard, P. (2024). Localizing Task Information for Improved Model Merging and Compression. arXiv preprint arXiv:2405.07813.
>
> [2] Ilharco, G., Ribeiro, M. T., Wortsman, M., Gururangan, S., Schmidt, L., Hajishirzi, H., & Farhadi, A. (2022). Editing models with task arithmetic. arXiv preprint arXiv:2212.04089.
>
> [3] Tang, A., Shen, L., Luo, Y., Xie, S., Hu, H., Zhang, L., ... & Tao, D. (2024). Smile: Zero-shot sparse mixture of low-rank experts construction from pre-trained foundation models. arXiv preprint arXiv:2408.10174.
>
> [4] Tang, A., Shen, L., Luo, Y., Yin, N., Zhang, L., & Tao, D. (2024). Merging Multi-Task Models via Weight-Ensembling Mixture of Experts. arXiv preprint arXiv:2402.00433.

---

### Meta-Review · Area_Chair_kGTW · 2024-12-20

**Metareview:**

The majority of the reviewers raised serious concerns regarding several aspects such as presentation clarify, methodological justification, empirical evaluation and contribution significance. The authors tried to response positively to those concerns, but failed to change the reviewer’s opinion. Overall, the paper may have potentials but need be improved a lot. This assertion also coincides with the author rebuttal, which indicates explicitly “To address your concern comprehensively, we will incorporate a task classifier to our method in the NEXT VERSION of our paper.”

**Additional Comments On Reviewer Discussion:**

The authors provided extensive rebuttal to try resolving reviewers’ concerns, implying that it is better to let the paper go through another round of peer review rather than accepting it at its current form.

---

### Decision · Program_Chairs · 2025-01-22

Reject